# Sulforaphane reduces obesity by reversing leptin resistance

**Işın Çakır[1,2], Pauline Lining Pan[1†], Colleen K Hadley[1,3], Abdulrahman El-Gamal[4], Amina Fadel[5], Dina Elsayegh[5], Omnia Mohamed[5], Nasser M Rizk[4,5*], Masoud Ghamari-Langroudi[2,6*]**

[1]Life Sciences Institute, University of Michigan, Ann Arbor, United States; [2]Department of Molecular Physiology & Biophysics, Vanderbilt University, Nashville, United States; [3]College of Literature, Science, and the Arts, University of Michigan, Ann Arbor, United States; [4]Biomedical Sciences Department, College of Health Sciences, Qu- Health, Qatar University, Doha, Qatar; [5]Biomedical Research Center, Qatar University, Doha, Qatar; [6]Warren Center for Neuroscience Drug Discovery, Department of Pharmacology, Vanderbilt University, Nashville, United States

**Abstract** The ascending prevalence of obesity in recent decades is commonly associated with soaring morbidity and mortality rates, resulting in increased health-care costs and decreased quality of life. A systemic state of stress characterized by low-grade inflammation and pathological formation of reactive oxygen species (ROS) usually manifests in obesity. The transcription factor nuclear factor erythroid-derived 2-like 2 (NRF2) is the master regulator of the redox homeostasis and plays a critical role in the resolution of inflammation. Here, we show that the natural isothiocyanate and potent NRF2 activator sulforaphane reverses diet-induced obesity through a predominantly, but not exclusively, NRF2-dependent mechanism that requires a functional leptin receptor signaling and hyperleptinemia. Sulforaphane does not reduce the body weight or food intake of lean mice but induces an anorectic response when coadministered with exogenous leptin. Leptin-deficient $Lep^{ob/ob}$ mice and leptin receptor mutant $Lepr^{db/db}$ mice display resistance to the weight-reducing effect of sulforaphane, supporting the conclusion that the antiobesity effect of sulforaphane requires functional leptin receptor signaling. Furthermore, our results suggest the skeletal muscle as the most notable site of action of sulforaphane whose peripheral NRF2 action signals to alleviate leptin resistance. Transcriptional profiling of six major metabolically relevant tissues highlights that sulforaphane suppresses fatty acid synthesis while promoting ribosome biogenesis, reducing ROS accumulation, and resolving inflammation, therefore representing a unique transcriptional program that leads to protection from obesity. Our findings argue for clinical evaluation of sulforaphane for weight loss and obesity-associated metabolic disorders.

**\*For correspondence:**
nassrizk@qu.edu.qa (NMR);
masoud.ghamari-langroudi@
vanderbilt.edu (MG-L)

**Present address:** †Department of Pharmacology, University of Michigan Medical Center, Ann Arbor, Michigan, United States

**Competing interest:** The authors declare that no competing interests exist.

## Introduction

There has been a sharp increase in the prevalence of obesity in the past decades. About 2 billion people in the world are either obese or overweight (*Finucane et al., 2011*; *National-Heart-Lung-Blood-Institute, 2013*; *Flegal et al., 2016*; *OECD, 2017*; *WHO, 2020*). Obesity commonly causes comorbidities, including diabetes, stroke, hypertension, cancer, liver failure, and arthritis. In fact, diabetes is becoming a pandemic disorder, and the most common underlying cause of diabetes is obesity. These comorbidities are associated with a measurable diminished quality of life and pose an enormous burden on health-care costs. Efforts to address the obesity epidemic pharmaceutically have been largely unsuccessful. To date, the most effective treatment available for obesity is the invasive

bariatric surgery, which is associated with short- and long-term complications. Therefore, development of pharmaceutical tools to tackle this pandemic is overdue.

The balance between food intake and energy expenditure (EE), that is, energy homeostasis, is regulated by central circuitries that integrate, among others, nutritional, hormonal, and visual cues to generate homeostatic or hedonic responses (*Andermann and Lowell, 2017*; *Rossi and Stuber, 2018*). Leptin-melanocortin circuitry is the major regulator of homeostatic feeding in mammals and represents a well-characterized system of peripheral-to-central communication of energy homeostasis. While lean or leptin-deficient animals are responsive to the anorectic and weight-reducing effect of leptin, diet-induced obesity manifests itself as a leptin-resistant state where hyperleptinemia is accompanied by lack of response to exogenous leptin administration (*Frederich et al., 1995*; *Myers et al., 2010*). Consequently, reversing leptin resistance or increasing leptin sensitivity has been a consideration to combat obesity (*Andreoli et al., 2019*). There are several mechanisms proposed in regard to the etiology of leptin resistance; however, whether the pharmacological approaches to potentiate leptin action can be effectively utilized therapeutically is not clear. For instance, earlier studies proposed that peripheral CB1R inverse agonism results in central leptin sensitization through cell-nonautonomous mechanisms, leading to a leptin-dependent weight loss in diet-induced obese (DIO) mice (*Tam et al., 2012*). Reported more recently, the central blockade of the glucose-dependent insulinotropic polypeptide receptor signaling reverses obesity in DIO mice but not in *Lep^{ob/ob}* mice, proposing a central and potentially cell-autonomous mechanism of leptin sensitization (*Kaneko et al., 2019*). Therefore, the origin and mechanism of action of signals that can alleviate leptin resistance seem diverse and largely unclear.

Obesity is usually accompanied by elevated markers of oxidative stress and low-grade systemic inflammation resulting, at least in part, from excess saturated free fatty acids (FAAs) and glucose (*Weisberg et al., 2003*; *Wellen and Hotamisligil, 2003*; *Manna and Jain, 2015*; *Ellulu et al., 2017*). FFAs act on TLR4 to initiate the expression of chemotactic pro-inflammatory signaling, including necrosis factor-alpha (TNF-α) and interleukin-6 (IL-6) (*Yeop Han et al., 2010*). TNF-α results in further lipolysis and activation of nuclear factor κB (NF-κB)/IKKβ pathway and increased phosphorylation of insulin receptor substrate 1 (IRS1). Signaling through the JAK-STAT3 pathway, IL-6 suppresses the expression of GLUT4 and IRS1 in skeletal muscle, resulting in further insulin resistance (*Kaul and Forman, 1996*; *Kahn and Flier, 2000*; *Qatanani and Lazar, 2007*; *Chen et al., 2015*). The insulin resistance decreases glucose uptake and increases FFA consumption in adipocytes, leading to increased β-oxidation in mitochondria. Overloading the electron flow to the mitochondrial electron transport chain results in excess reactive oxygen species (ROS) generation, leading to further activation of pro-inflammatory molecules (*Kaul and Forman, 1996*). The ROS accumulation and inflammatory response are further exacerbated by eventual suppression of β-oxidation, and the resulting reduction of nicotinamide adenine dinucleotide phosphate (NADPH) oxidase 4 (NOX4) activity (*Lugrin et al., 2014*). Therefore, interventions that increase mitochondrial biogenesis and beta-oxidation enzymes can potentially diminish ROS production and alleviate the inflammation and insulin resistance accompanying obesity.

The nuclear factor erythroid-derived 2-like 2 (NRF2) is the master transcriptional regulator of cytoprotective pathways involving detoxification, antioxidation, metabolism, and inflammation (*Vomund et al., 2017*). Accumulation of ROS and depletion of antioxidant capacity activate NRF2 (*Diotallevi et al., 2017*). Under normal conditions, NRF2 is sequestered in the cytosol by Kelch-like ECH-associated protein 1 (KEAP1), which recruits NRF2 for the proteasome-mediated degradation (*McMahon et al., 2003*). Molecules that interact with the reactive cysteine residues on KEAP1, such as ROS, set NRF2 free to translocate into the nucleus. NRF2 activation induces expression of antioxidative genes, including NAD(P)H quinone oxidoreductase 1, glutathione S-transferases, and glutamate cysteine ligase, which subsequently reduce ROS (*McWalter et al., 2004*). ROS formation also triggers inflammatory responses, which lead to activation of IκB kinase (IKK) β and phosphorylation of IκBα, resulting in nuclear translocation of the transcription factor NF-κB, the master regulator of the inflammatory signals (*Napetschnig and Wu, 2013*; *Wenzel et al., 2017*).

Similar to NRF2, IKKβ contains a motif allowing KEAP1 binding. Under basal conditions, KEAP1-dependent degradation of IKKβ results in IκBα inhibition of NF-κB activation. In response to pathological ROS formation, KEAP1 is inhibited, resulting in IKKβ stabilization, and subsequent phosphorylation and degradation of IκBα. The absence of IκBα inhibition on NF-κB permits its activation and nuclear

translocation (*Lee et al., 2009*). Therefore, the antioxidant role of NRF2 also helps in the resolution of inflammation (*Wenzel et al., 2017*; *Cuadrado et al., 2018*).

The role of NRF2 in obesity and diabetes has been controversial. Global deletion of NRF2 in mice on high-fat diet (HFD) results in lower body weight compared to the wild-type counterparts (*Pi et al., 2010*) and ameliorates HFD-induced insulin resistance by elevating systemic FGF21 levels (*Chartoumpekis et al., 2011*; *Zhang et al., 2012*). However, activation of NRF2 is also protective against pathological ROS formation in insulin-sensitive tissues and prevents exacerbated diabetic complications such as nephropathy, retinopathy, and endothelial dysfunction (*Yoh et al., 2008*; *Xu et al., 2014*; *da Costa et al., 2019*). Accordingly, several studies using pharmacological NRF2 activators suggested a protective role of NRF2 activation against obesity and insulin resistance (*Yu et al., 2011*; *He et al., 2012*; *Bahadoran et al., 2013*). Indirect genetic activation of NRF2 achieved by KEAP1 deletion confers protection from short-term HFD-induced metabolic dysfunctions, while chronic long-term NRF2 activation results in increased weight gain and elevated markers of the metabolic syndrome (*More et al., 2013*). Furthermore, activation of NRF2 restricted only to muscle can lead to resistance to obesity and diabetes (*Uruno et al., 2016*; *Matzinger et al., 2018*), suggesting that metabolic consequences of NRF2 activation by genetics versus pharmacological tools differ across the tissues.

Sulforaphane (SFN) is a natural isothiocyanate produced by cruciferous vegetables such as broccoli, Brussels sprouts, and cabbage. SFN interacts with the active cysteine residues of KEAP1, leading to the dissociation and activation of NRF2 (*Fahey and Talalay, 1999*; *Morimitsu et al., 2002*; *de Figueiredo et al., 2015*). SFN has a cytoprotective role by triggering oxidative stress response (OSR) and alleviating inflammation in multiple models of cellular stress. By suppression of the NF-κB pathway, SFN protects against damages in diabetic neuropathy and high glucose-induced changes in rodents (*Song et al., 2009*; *Negi et al., 2011*). SFN induces protection against diabetic cardiomyopathy by upregulation of NRF2 (*Bai et al., 2013*) and reversing the oxidative stress-induced inhibition of LKB1/AMPK pathway (*Zhang et al., 2014*). SFN alleviates alcohol-induced oxidative stress and endoplasm reticulum stress in mice (*Lei et al., 2018*) and mitochondrial dysfunction in nonalcoholic fatty liver disease (*Xu et al., 2019*). By suppressing the expression of adipogenic factors peroxisome proliferator-activated receptor γ (PPARγ) and CCAAT/enhancer-binding protein α (C/EBPα), SFN blocks adipocyte differentiation and therefore decreases lipid accumulation (*Choi et al., 2012*; *Choi et al., 2014*). Treatment of mice with SFN was further reported to result in AMPK activation (*Choi et al., 2014*), suggesting that SFN can have antiobesity and antidiabetic actions. Accordingly, SFN supplementation to obese patients with dysregulated type 2 diabetes reduces fasting blood glucose and glycated hemoglobin (HbA1c), with its antidiabetic properties comparable to the most widely prescribed antidiabetic medication metformin (*Hawley et al., 2002*; *Axelsson et al., 2017*). SFN also ameliorates glucose intolerance in obese mice by upregulation of the insulin signaling pathway (*Xu et al., 2018b*). As a pleiotropic molecule, SFN targets other homeostatic pathways besides NRF2. For example, SFN acts as an HSP90 inhibitor, and HSP90 inhibition was proposed to protect mice from HFD-induced weight gain (*Chartoumpekis et al., 2011*; *Li et al., 2012*) and regulate gene expression by increasing histone acetylation through HDAC inhibition (*Kim et al., 2017*).

Previous studies have separately addressed the distinct roles of NRF2 and leptin signaling on energy homeostasis and metabolic syndrome (*Shawky and Segar, 2018*). However, whether the two pathways interact, especially in a cell-nonautonomous manner, is unknown. Using a combination of methods of pharmacology and various mouse models of obesity, here we explore the role of SFN as a molecule at the intersection of the NRF2 pathway and central leptin signaling.

## Results
### SFN induces weight loss in diet-induced obese mice

In order to assess the potential role of SFN on energy metabolism, we first tested its effect in obese mice. We fed lean wild-type, C57BL/6J mice a HFD (60% of the calories from fat) for 16–20 weeks to induce obesity. In order to test the potential antiobesity effect of SFN, we treated these DIO mice by daily intraperitoneal SFN (i.p., 5 mg/kg) injections. During the 2-week treatment period, SFN treatment led to a significant weight loss in mice (4.8% ± 1.4% vs. 18.2% ± 1.2% of initial weight, vehicle vs. SFN, p<0.0001) (*Figure 1A and B*). The reduced body weight was accompanied by a significant decrease in the food intake compared to the vehicle-treated counterparts (week 1: 1.74 ± 0.06 g

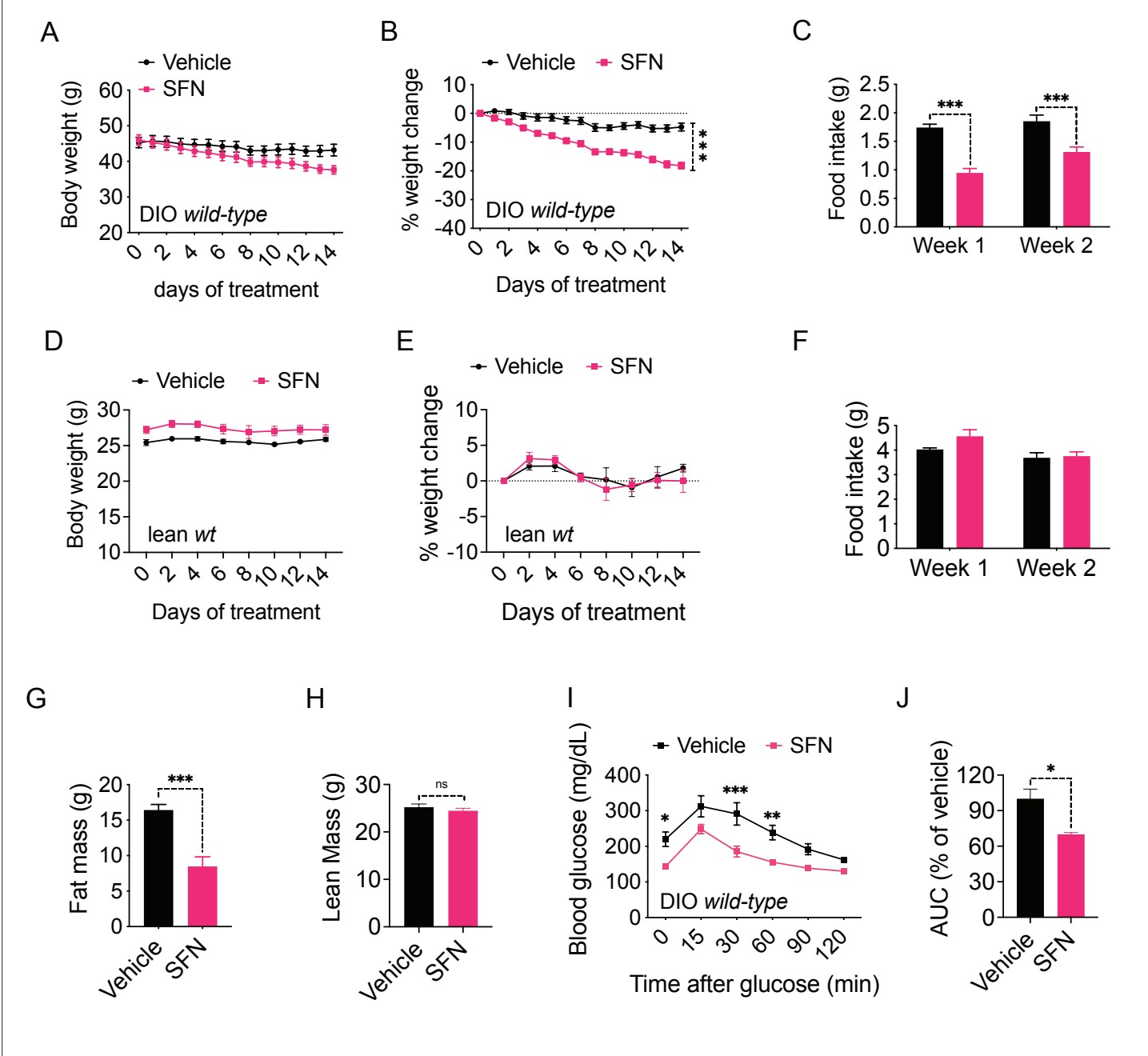

**Figure 1.** Sulforaphane (SFN) reduces diet-induced obesity. (**A–C**) Diet-induced obese (DIO) (n = 7–8) or (**D–F**) lean wild-type mice (n = 4) were treated by daily i.p. vehicle or SFN (5 mg/kg) administrations. (**A, D**) Body weight, (**B, E**) percent change in body weight, and (**C, F**) average daily food intake of the mice measured at indicated times. (**G, H**) Fat mass (**G**) and lean mass (**H**) of the DIO mice (n = 6–7) after SFN treatment. (**I, J**) Glucose tolerance test (GTT) (**I**) and the area under the curve (AUC) of the GTT (**J**) performed on DIO mice (n = 4) following 2 weeks of vehicle or SFN treatment. GTT was conducted after an overnight fast. *p<0.05, **p<0.01, ***p<0.001 by two-way ANOVA with Sidak correction (**A–F, I**), or Student's *t*-test (**G, H, J**).

The online version of this article includes the following figure supplement(s) for figure 1:

**Figure supplement 1.** Sulforaphane (SFN) induces weight loss in obese CD1 mice.

vs. 0.95 ± 0.08 g, p<0.0001; week 2: 1.85 ± 0.11 g vs. 1.31 ± 0.09 g, p<0.001; vehicle vs. SFN) (*Figure 1C*). We next tested if the weight loss-inducing effect of SFN required obesity. To this end, we treated lean wild-type mice with the same dose of SFN (i.p., 5 mg/kg) daily and monitored their body weight and food intake. In contrast to its effect on DIO mice, SFN did not alter the weight or food intake of the lean mice (*Figure 1D–F*). In order to ensure that the effect of SFN was not restricted to

the specific C57BL/6J strain of mice, we repeated these experiments on the CD1 strain. During the 30-day treatment, SFN reduced the body weight and cumulative food intake (CFI) of DIO CD1 mice as effectively (body weight change was +2.05 ± 0.99 g vs. –8.25 ± 1.19 g for vehicle vs. SFN, respectively, p<0.0001) (*Figure 1—figure supplement 1A–C*). These results collectively suggest that SFN is an antiobesity molecule and alters feeding and body weight selectively in obese mice.

We next analyzed the change in the body composition of the DIO C57BL/6J mice after 2 weeks of SFN treatment as described above. NMR-based quantification of the lean and fat mass showed that SFN caused a significant decrease in fat depots (total fat mass: 16.43 ± 0.77 g vs. 8.47 ± 1.36 g, vehicle vs. SFN, respectively, p<0.001) while the lean mass of the animals was not affected (*Figure 1G and H*), suggesting that SFN-induced weight loss was selectively due to decreased adiposity. Excess fat accumulation and thus obesity is the leading risk factor for dysregulation of glucose homeostasis and diabetes. Consistent with their decreased adiposity, SFN-treated DIO mice performed better during an i.p. glucose tolerance test (GTT) after an overnight fast (*Figure 1I and J*) and after a 6 hr daytime fast (*Figure 1—figure supplement 1D–G*). The fasting glucose normalized to body weight was also lower in the SFN-treated group (*Figure 1—figure supplement 1H*). This effect was limited to obese mice such that SFN did not alter the glucose tolerance of lean mice (*Figure 1—figure supplement 1I and J*).

## SFN improves metabolic function in obese mice

In order to analyze the SFN-induced changes in energy metabolism in more detail, we placed the DIO mice into the Promethion systems metabolic chambers after acclimation and started treating them daily with either vehicle or SFN. During this 1-week treatment period, SFN significantly suppressed the food intake during both light and dark cycles (*Figure 2A*). Despite the decreased caloric intake, the total EE of the mice did not show any significant changes (*Figure 2B*), suggesting that SFN prevents the calorie restriction-induced suppression of EE and may activate mechanisms that increase EE. The respiratory quotient (also called the respiratory exchange ratio [RER]) of the SFN-treated mice was significantly lower in most of the measured cycles (*Figure 2C*). Decreased RER is an indication of increased oxidation of fatty acids as opposed to carbohydrates as the energy source, thus SFN-induced decrease in RER is consistent with the decreased fat mass of the SFN-treated animals (*Figure 1G*). Accordingly, SFN administration downregulated the expression of the fatty acid synthesis genes in the liver and white adipose tissues of the DIO mice (*Figure 2D and E*). The physical activity of the SFN-treated animals was not significantly different compared to the vehicle-treated counterparts (*Figure 2F–H*), suggesting that SFN does not induce any apparent toxicity or aversive effects.

Our results indicate that SFN induces weight loss in obese mice by suppressing food intake and blocking fatty acid synthesis while promoting fat oxidation, with no obvious effects in lean mice. We next tested whether SFN would protect the lean mice against diet-induced weight gain when exposed to HFD. To this end, we placed lean wild-type mice on HFD and treated them with daily injections of vehicle or SFN (10 mg/kg). Another group of mice were kept on standard chow diet and treated with vehicle in a similar manner as control. We assessed the effect of SFN treatment on development of diet-induced obesity in mice for eight consecutive weeks. At the end of this period, vehicle-treated animals on HFD gained significant adiposity with their final weight reaching 60.56% ± 2.50% of their initial body weights compared to the vehicle-treated mice on standard chow diet that gained 14.0% ± 1.35% weight. SFN protected the mice against diet-induced obesity, as the SFN-treated mice presented with an intermediate body weight gain of 42.64% ± 3.47% after the 8-week treatment period (*Figure 3A and B*). The lean mass of the animals was not significantly different at the end of the experiment; however, their final fat mass was significantly different, indicating that SFN significantly blocked the HFD-induced adipose tissue accumulation (13.19 ± 0.95 g in HFD-vehicle vs. 8.34 ± 0.84 g in HFD-SFN, p<0.001, *Figure 3C*). The CFI of the animals on HFD was not different between the HFD-vehicle vs. HFD-SFN groups during this period, and the mice on standard chow diet consumed less calories overall (*Figure 3D*). Repeating the experiment with a slightly higher dose of SFN (15 mg/kg) led to a significantly decreased calorie intake and attenuation of weight gain on HFD (*Figure 1—figure supplement 1K and L*), suggesting that SFN attenuates weight gain and food intake in a dose-dependent manner. Consistent with their lower fat mass, SFN-treated mice had lower plasma leptin levels (*Figure 3E*) and displayed significantly lower blood glucose (*Figure 3F*). Taken collectively, SFN acts to prevent the development of obesity and associated hyperglycemia.

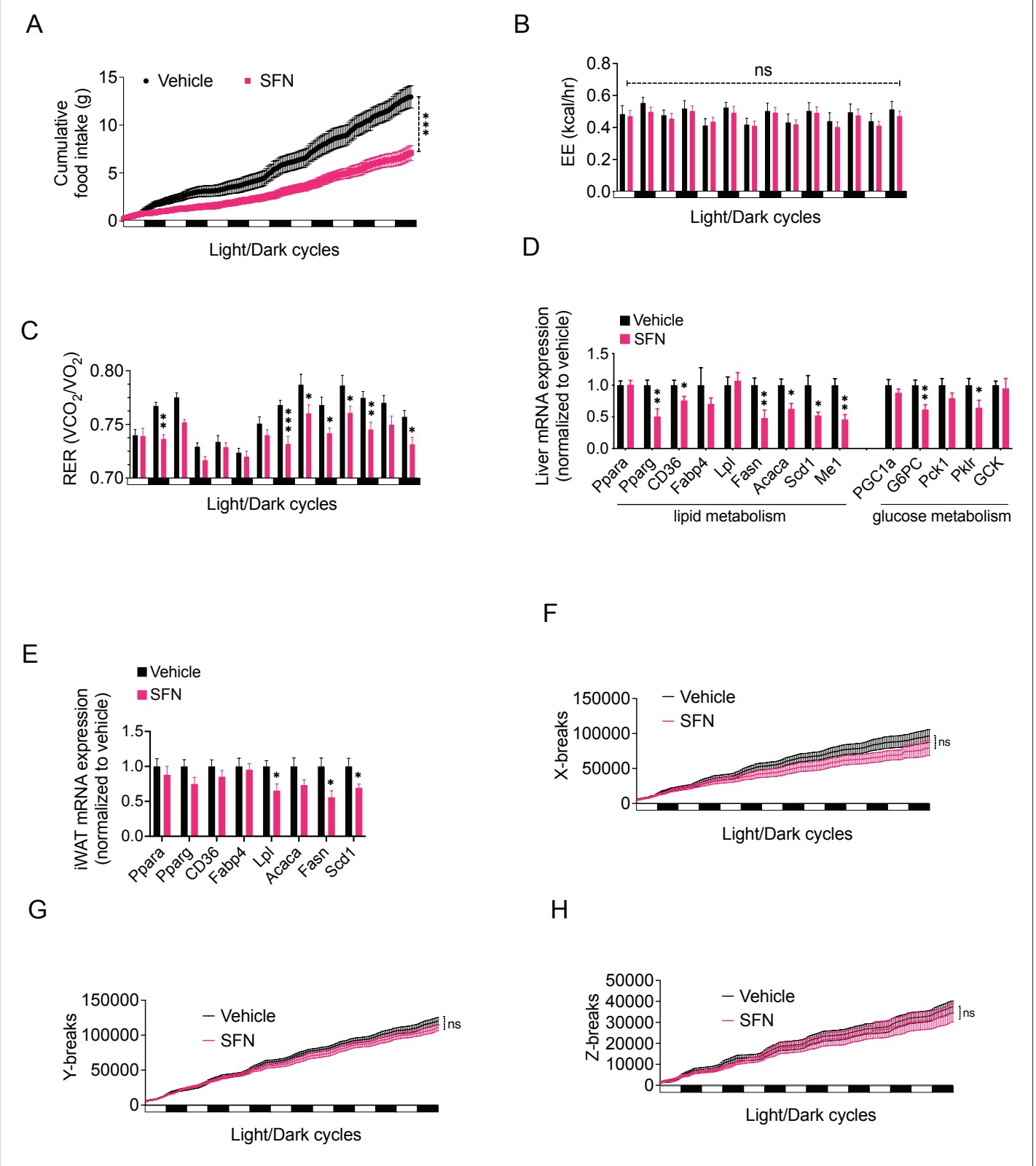

**Figure 2.** Sulforaphane (SFN) improves metabolic function in diet-induced obese (DIO) mice. (**A–C, F–H**) SFN's effect on the parameters of metabolism in wild-type DIO mice. Mice were acclimated to single housing and daily handling for 1 week, and i.p. injections of saline for 3 days prior to being placed in metabolic cages. After acclimation, DIO wild-type mice (n = 6–8) were placed into Promethion systems, SABLE, metabolic chambers and started receiving vehicle or 5 mg/kg SFN injections daily after being in the chambers. Cumulative food intake (**A**), energy expenditure (EE)

*Figure 2 continued on next page*

*Figure 2 continued*

(**B**), respiratory exchange ratio (RER) (**C**), and physical activity on the three axes (**F–H**) were recorded for light and dark cycles. (**D, E**) Analysis of gene expression by qPCR in liver (**D**) and inguinal white adipose tissue (WAT) (**E**) of DIO mice (n = 7) following treatment with vehicle or SFN. *p<0.05, **p<0.01, ***p<0.001 by two-way ANOVA with Sidak correction (**A–C, F–H**), or Student's *t*-test (**D, E**).

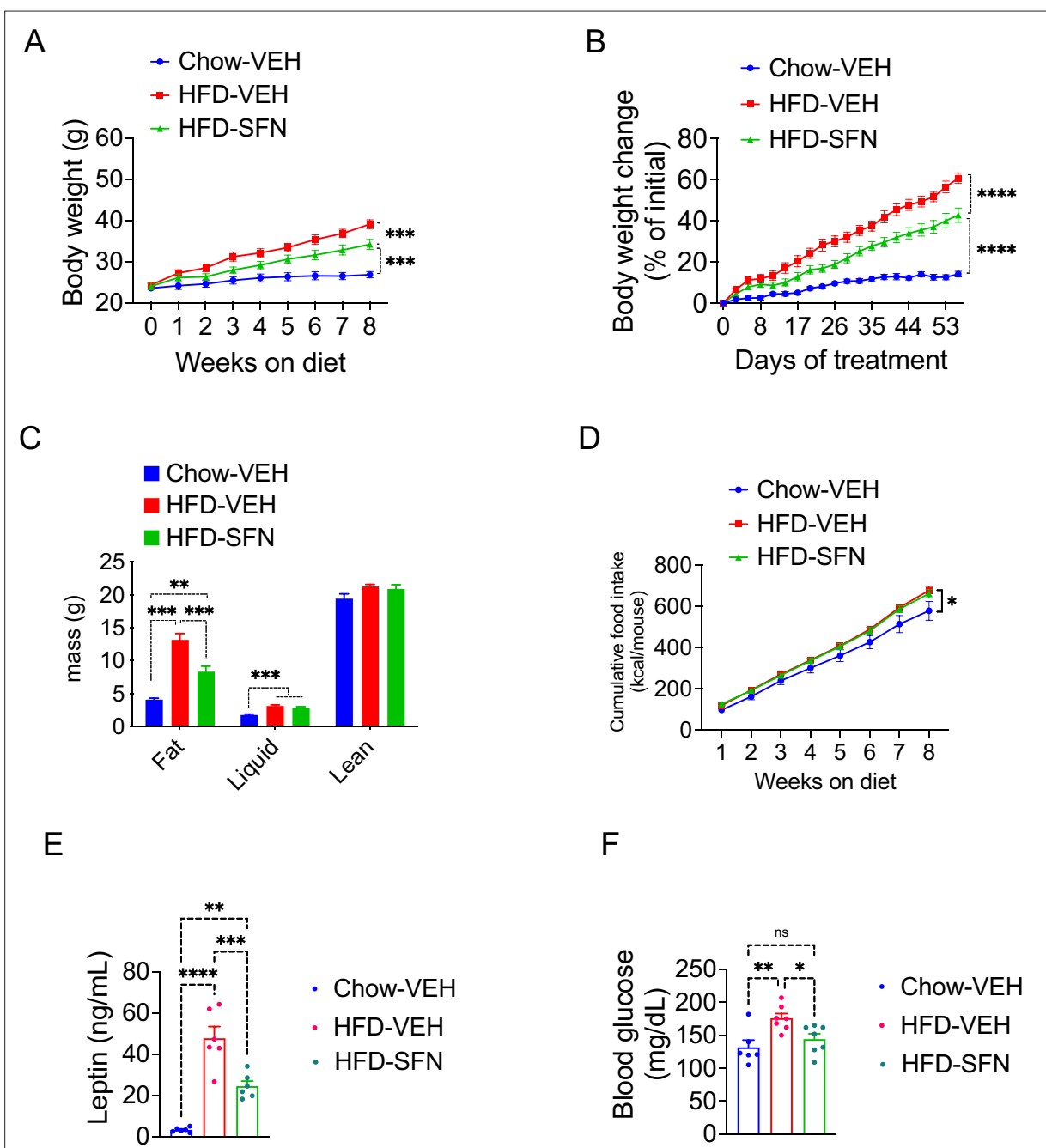

**Figure 3.** Sulforaphane (SFN) prevents the development of diet-induced obesity. Lean wild-type mice (n = 6–7 per group) were treated with vehicle on standard chow (Chow) or placed on high-fat diet (HFD) and treated daily with either vehicle or SFN (i.p., 10 mg/kg) for eight consecutive weeks. (**A**) Body weight and (**B**) percent change in body weight of the animals during the treatment period. (**C**) Body composition of the mice at the end of the treatment period. (**D**) Cumulative food intake during the treatment period. (**E, F**) Plasma leptin (n = 6–10) (**E**) and blood glucose levels (n = 6–14) (**F**) measured at the end of 8-week study. *p<0.05, **p<0.01, ***p<0.001 by two-way ANOVA with Tukey's post-hoc test (**A, B, D**), one-way ANOVA (**C, E, F**).

## SFN action requires NRF2

SFN is a pleiotropic compound with various biological targets reported (*Malavolta et al., 2018*). For example, SFN is reported to inhibit the activity of histone deacetylase 6 (HDAC6) (*Gibbs et al., 2009*; *Ho et al., 2009*), HSP90, and mTOR (*Bali et al., 2005*; *Kovacs et al., 2005*; *Zhang et al., 2021*). Most notably, through its direct interaction with KEAP1, SFN is well characterized to lead to NRF2 activation. Therefore, we tested whether the antiobesity effect of SFN was NRF2 dependent. NRF2 knockout (KO) mice were fed HFD for 15 weeks to induce obesity, weighing at least 45 g. Next, we treated the mice with daily i.p. injections of either vehicle or SFN (5 mg/kg) for three consecutive weeks.

SFN did not induce any significant changes in body weight of DIO nrf KO mice during the first 2 weeks of SFN administration. During the third week, however, SFN treatment led to a small but significant weight loss in these mice (*Figure 4A and B*). Doubling the administered SFN dose to 10 mg/kg after 3 weeks did not result in further body weight change. At the end of the treatment period, the SFN group had lower absolute body weight (47.6 ± 3.5 g in vehicle ss. 45.7 ± 2.1 g in SFN, p=0.58) and lower percentage of initial body weight (2.5% ± 2.8% vs. –2.5 ± 2.5% of initial weight, vehicle vs. SFN, p=0.0032) than the control mice (*Figure 4A and B*). Comparison of wild-type DIO and NRF2 KO DIO cohorts indicates that SFN-induced weight loss is much smaller and with significant delay in NRF2 KO animals (*Figure 4C and D*). The SFN-treated NRF2 KO DIO mice displayed a tendency for decreased daily food intake (*Figure 4E*); however, this trend did not reach statistical significance. We did not detect a significant decrease in caloric intake when the food intakes were analyzed cumulatively (*Figure 4F*), suggesting that SFN induces weight loss predominantly, but not exclusively, in an NRF2-dependent manner. These results are in agreement with a previous report demonstrating an SFN-induced NRF2-independent pathway (*Nagata et al., 2017*). SFN-treated NRF2 KO DIO mice failed to perform better during an i.p. GTT after a 6 hr daytime fast (*Figure 4G and H*). SFN was reported to have HDAC6 inhibitory activity (*Gibbs et al., 2009*), and HDAC6 inhibitors induce weight loss in DIO mice by suppressing calorie intake (unpublished observations). SFN induced weight loss in HDAC6 KO obese mice in a comparable extent to the wild-type counterparts (*Figure 4I and J*). The SFN-induced suppression of food intake was also not different between the wild-type and HDAC6 KO mice (*Figure 4K*). Furthermore, we did not detect an HDAC6 inhibitory activity of SFN in a cell-based assay (*Figure 4—figure supplement 1*). Analysis of the SFN-induced weight loss from DIO wild-type, NRF2 KO, and HDAC6 KO mice suggests that NRF2, but not HDAC6, is required for the antiobesity effect of SFN.

## SFN reverses obesity by increasing leptin action

SFN was proposed to confer protection from obesity, at least in part, by inducing adipose tissue browning (*Nagata et al., 2017*). We thus tested if SFN-treated mice had elevated markers of adipose tissue browning or beiging in various fat depots. Analysis of gene expression in inguinal white adipose tissue (iWAT) or brown adipose tissue (BAT) of wild-type DIO mice failed to detect a significant increase in the expression of markers of adipose tissue browning or beiging (*Figure 4—figure supplement 2A and B*). Surprisingly, SFN induced a significant increase in the expression of BAT-marker genes, such as *Ucp1*, in the iWAT of NRF2 KO mice (*Figure 4—figure supplement 2C and F*), potentially accounting for the small but significant weight loss effect observed in the NRF2 KO animals (*Figure 4A and B*). Treatment of wild-type iWAT explants with SFN did not alter the expression of either beige or brown adipose-specific genes (*Figure 4—figure supplement 2G and H*), while the NRF2 target gene NQO1 expression was significantly upregulated (*Figure 4—figure supplement 2I*), leading us to conclude that increased adipose tissue beiging or browning and associated fat thermogenesis are independent of the SFN-induced weight loss.

Our results indicate that daily SFN administration caused weight loss in DIO mice but not in standard chow-fed lean mice, hence supporting the model that the weight-reducing effect of SFN requires hyperleptinemia. We thus hypothesized that the effect of SFN is compromised in mice with disrupted leptin receptor signaling. We tested this hypothesis by examining the antiobesity effect of SFN on two established genetic models of obesity, namely, the leptin-deficient *Lep^{ob/ob}* and the leptin receptor mutant *Lepr^{db/db}* mice, both of which develop early-onset hyperphagic obesity due to deficient leptin signaling. To this end, we treated *Lepr^{db/db}* mice with daily i.p. SFN injections for 4 weeks. During the course of treatment, *Lepr^{db/db}* mice did not lose body weight and continued to consume comparable

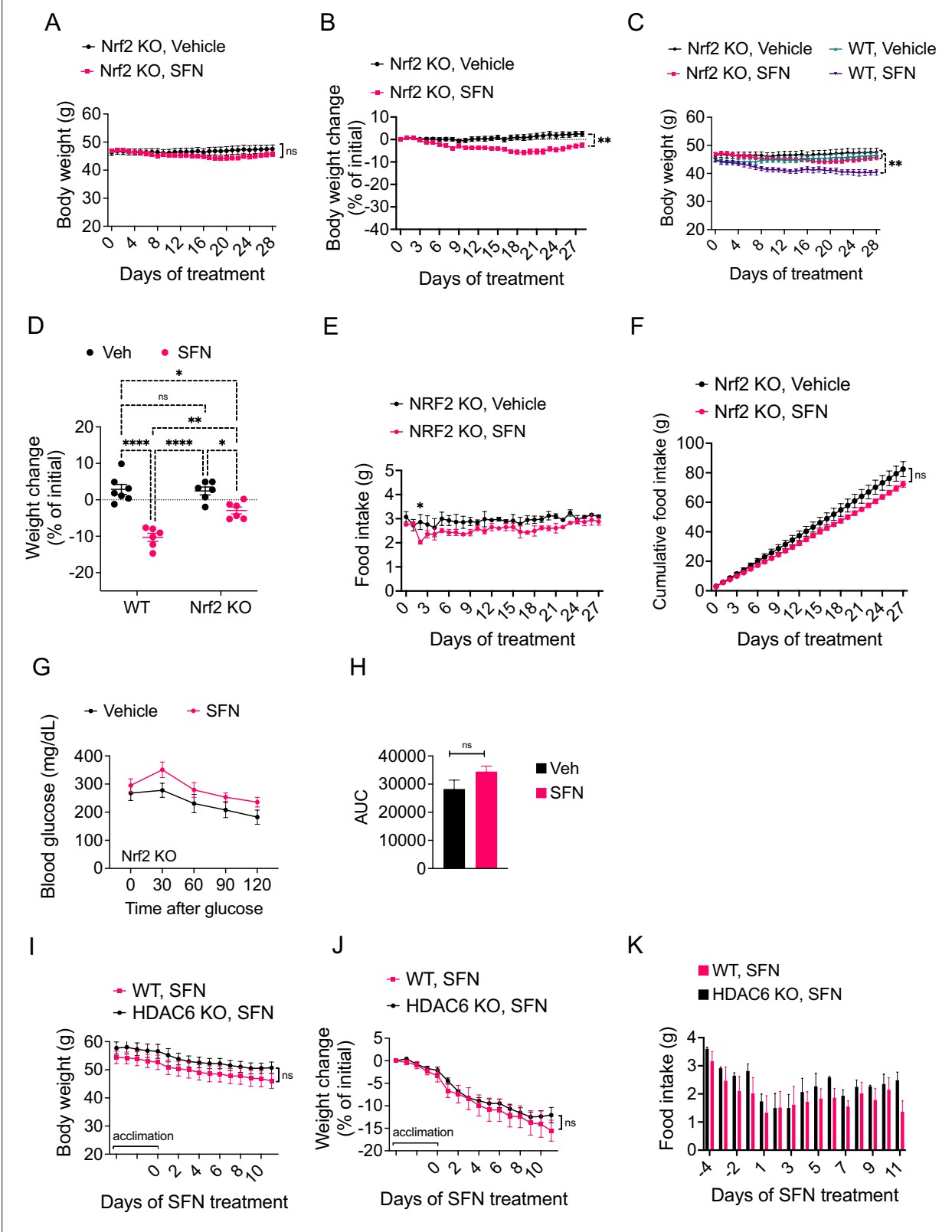

**Figure 4.** The antiobesity effect of sulforaphane (SFN) is NRF2-dependent and HDAC6-independent. (**A–H**) NRF2 KO mice were placed on high-fat diet (HFD) to induce obesity. Mice were then treated daily by i.p. vehicle or SFN injections. SFN dose was 5 mg/kg for the first three weeks and increased to 10 mg/kg thereafter. Wild-type diet-induced obese (DIO) mice were treated with vehicle or SFN for the same period for comparison. (**A, B**) Body weight (**A**) and percent change in body weight (**B**) of the NRF2 KO mice throughout the vehicle or SFN treatments. (**C**) Body weight of NRF2 KO (the same

*Figure 4 continued on next page*

*Figure 4 continued*

cohort graphed in **A**) and wild-type DIO mice receiving treatments with vehicle or SFN for the same period for comparison. (**D**) Percent change in body weight of NRF2 KO and wild-type DIO mice at the end of the treatments shown in panel (**C**). (**E, F**) Daily food intake (**E**) and cumulative food intake (**F**) of the DIO NRF2 KO mice (n = 6 per group) during the treatment period. (**G, H**) Glucose tolerance test (GTT) (**G**) and the area under the curve (AUC) of the GTT (**H**) performed on NRF2 KO DIO mice (n = 6, 8) during the 3 weeks of vehicle or 5 mg/kg SFN treatment shown in (**A**). (**I–K**) HDAC6 KO or wild-type mice were fed HFD to induce obesity. The mice were then treated with vehicle or 5 mg/kg SFN by daily i.p. injections. Body weights (**I**), percent change in body weight (**J**), and daily food intakes (**K**) were recorded (n = 4–6 mice per group). *p<0.05, **p<0.01, ***p<0.001 by two-way ANOVA with Sidak correction (**A–I**).

The online version of this article includes the following figure supplement(s) for figure 4:

**Figure supplement 1.** Effect of sulforaphane (SFN) on tubulin acetylation in mouse embryonic fibroblasts (MEFs).

**Figure supplement 2.** Sulforaphane (SFN) does not induce a beige or brown adipose-specific gene expression profile.

calories to the vehicle-treated counterparts, suggesting that the antiobesity effect of SFN was blunted in leptin receptor mutant *Lepr^{db/db}* mice (*Figure 5A–C*). We next treated *Lep^{ob/ob}* mice with vehicle or SFN and observed that *Lep^{ob/ob}* mice also continued their normal weight gain and energy consumption during the period of SFN administration (*Figure 5D–F*). Notably, *Lepr^{db/db}* or *Lep^{ob/ob}* mice did not have improved glucose homeostasis during the SFN treatment as assessed by GTT (*Figure 5—figure supplement 1A and B*), suggesting that SFN improves the glucose metabolism (*Figure 1I and J*) mainly by weight reduction. These results collectively led us to conclude that the antiobesity effect of SFN required a functional leptin receptor signaling. Therefore, we tested whether SFN can lead to weight loss if the leptin deficiency in *Lep^{ob/ob}* mice is restored. *Lep^{ob/ob}* mice have functional leptin receptors and are very leptin sensitive (*Coleman, 1973*; *Baskin et al., 1998*). Thus, in order to assess if SFN could potentiate the weight-reducing effect of leptin, following 3 days of vehicle or SFN administration, we started co-treating the *Lep^{ob/ob}* animals with 0.2 mg/kg leptin daily for the next 6 days (*Figure 5G–I*). While SFN alone did not alter the body weight or food intake of the *Lep^{ob/ob}* mice, when it was administered in the presence of leptin, SFN led to a pronounced decrease in the food intake and body weights of the animals compared to leptin alone (*Figure 5G–I*), suggesting that the antiobesity effect of SFN required leptin.

DIO mice develop hyperleptinemia due to excess adipose tissue accumulation; however, they do not respond to the anorectic effect of administered leptin due to obesity-associated leptin resistance (*Enriori et al., 2007*). Since the metabolic action of SFN requires high levels of leptin, we hypothesized that SFN induces weight loss in leptin-resistant DIO wild-type mice by increasing leptin sensitivity. In support of this hypothesis, lean wild-type mice with low circulating leptin concentrations fail to respond to weight-reducing effects of SFN treatment. We thus examined whether SFN can potentiate leptin action in lean wild-type mice when serum leptin levels are exogenously elevated. We treated fasted lean wild-type mice with either vehicle or SFN in combination with saline or leptin and monitored the food intake and weight changes in the animals during the refeeding period. As expected, leptin + vehicle attenuated the weight gain and food intake of the lean mice (*Figure 5J and K*), while the effects of saline + SFN and saline + vehicle on food intake and weight changes were not different. Notably, leptin + SFN-treated mice ate significantly less food and gained less weight than leptin + vehicle-treated animals (*Figure 5J and K*), suggesting that SFN increases the anorectic action of leptin.

## Skeletal muscle serves as the primary site of SFN action

Our results indicate that SFN confers protection against diet-induced obesity in an NRF2 and leptin-dependent manner. Leptin receptors are expressed throughout the body; however, the functional isoform, called LepRb, is predominantly expressed in central feeding centers, most notably in the hypothalamus. The arcuate nucleus of the hypothalamus (ARC) harbors GABAergic LepRb-positive neuronal populations that appear to be the key regulators of homeostatic feeding (*Baskin et al., 1999*; *Zhu et al., 2020*). The leptin resistance has also been predominantly mapped to the central nervous system (*Xu et al., 2018a*). Therefore, we next tested if SFN reduces obesity through its direct central action. Central administration of 5–10 µg of SFN for 5 days failed to reduce body weight or food intake of DIO mice (*Figure 5—figure supplement 2A and B*). Furthermore, SFN failed to enhance leptin receptor signaling in LepRb-expressing hypothalamic N1 cells (*Figure 5—figure supplement 2C*), suggesting that the leptin-sensitizing effect of SFN was not LepRb^+-cell autonomous.

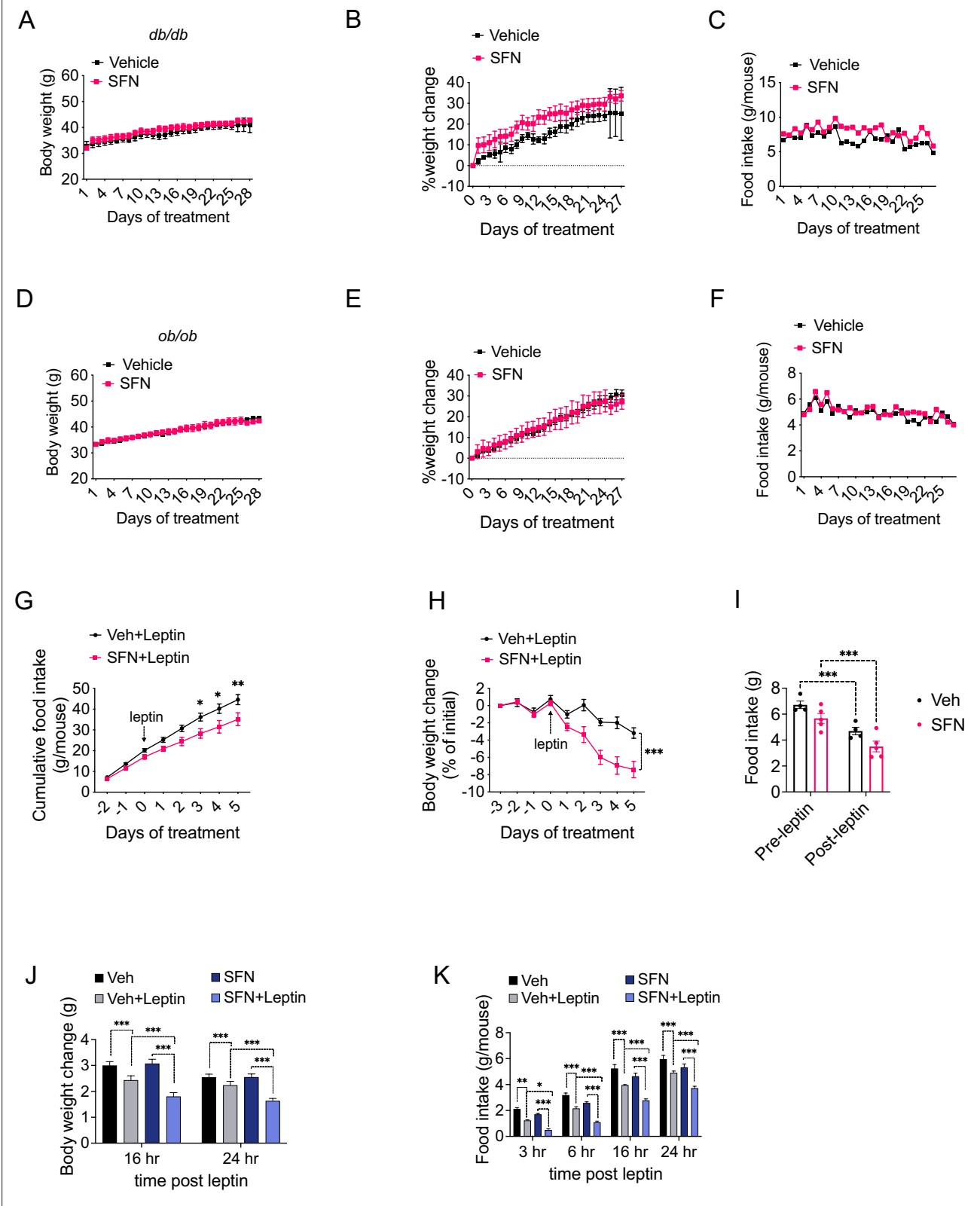

**Figure 5.** Sulforaphane (SFN) reduces obesity by improving leptin action. (**A–F**) Leptin signaling is required for SFN's action. *Lepr*^db/db^ mice (n = 3 per group) (**A–C**) and *Lep*^ob/ob^ mice (n = 3 per group) (**D–F**) fed with normal chow diet were treated with vehicle or 5 mg/kg SFN for 4 weeks by daily i.p. injections. Body weight (**A, D**), percent change in body weights (**B, E**), and food intake (**C, F**) of the animals. (**G–I**) Cumulative food intake (**G**), body weight change (percent of initial) (**H**), and average daily food intakes (before and after leptin injections, **I**) of *Lep*^ob/ob^ mice (n = 6 mice per group)

*Figure 5 continued*

pretreated for 3 days with vehicle or 5 mg/kg SFN, then started receiving daily i.p. injections of leptin (0.2 mg/kg). (**J, K**) Post 24 hr fast-refeeding kinetics in normal chow-fed lean wild-type mice (n = 15–16 mice per group) show the body weight change (**J**) and food intake (**K**) of mice that received single dose of vehicle, saline, 5 mg/kg SFN and/or 0.2 mg/kg leptin 1 hr before refeeding begins. *p<0.05, **p<0.01, ***p<0.001 by two-way ANOVA with Sidak correction (**A, B, D, E, G–I**), or Tukey's post-hoc test (**J, K**).

The online version of this article includes the following figure supplement(s) for figure 5:

**Figure supplement 1.** Sulforaphane (SFN) requires leptin action for improved glucose homeostasis.

**Figure supplement 2.** The antiobesity effect of sulforaphane (SFN) is not centrally mediated.

**Figure supplement 3.** Sulforaphane (SFN) acts primarily in the periphery.

Thus, we next probed the NRF2 activity in several metabolic tissues including the hypothalamus, liver, eWAT, iWAT, and skeletal muscle following peripheral SFN administration. Because *Hmox1* was the most upregulated OSR gene among others tested in the SFN-treated mouse embryonic fibroblasts (*Figure 5—figure supplement 3A*), we used the HMOX1 expression as a surrogate for NRF2 activation (*Wagner et al., 2015*).

Surprisingly, SFN treatment resulted in upregulation of HMOX1 expression only in the skeletal muscle but not in the other tissues that we analyzed (*Figure 6A*). This effect was NRF2 dependent as SFN failed to upregulate *Hmox1* expression in NRF2 KO mice (*Figure 6B*). These results suggested that the skeletal muscle is potentially a direct target tissue of SFN action. To understand the global transcriptional changes following SFN treatment, we conducted RNAseq on tissues from DIO wild-type mice following 1-week treatment with vehicle or SFN (*Supplementary file 1*). We also analyzed tissues from DIO NRF2 KO mouse tissues following vehicle or SFN treatments for comparison (*Supplementary file 2*). As tissues of interest, we focused our analysis on the adipose tissue depots (iWAT, eWAT, and BAT), liver, skeletal muscle, and the hypothalamus. In the wild-type mice, the highest transcriptional changes were detected in the liver, and the least number of transcriptional changes were observed in the hypothalamus (*Figure 5—figure supplement 3B*, *Figure 6—figure supplement 1*), which might be explained by the highly heterogenous nature of the hypothalamus. This is also in agreement with LepRb-positive cells representing only a small population of the hypothalamus (*Allison et al., 2018*). In NRF2 KO mice, the number of SFN-induced transcriptional changes were significantly reduced compared to the wild-type counterparts (*Figure 5—figure supplement 3B*). While SFN treatment altered the expression of 2555 genes in the liver of wild-type mice, only 77 genes were significantly regulated in the NRF2 KO mouse liver, with similar trends of lower number of differentially expressed genes detected in tissues from KO mice (*Figure 5—figure supplement 3B*).

## Validating SFN site of action by using RNAseq and in silico analysis

SFN treatment leads to a significant reduction in food intake and body weight (*Figure 1*); therefore, we thought that the transcriptional output we observe is likely a combination of direct SFN action in the respective tissues and the changes induced secondary to the metabolic phenotype induced by SFN. Therefore, we blasted the transcriptional profile of individual tissues in the Connectivity Map (CMap) database (*Lamb et al., 2006*). CMap contains gene expression profiles of cell lines subjected to genetic perturbation (gain and loss of function) or chemical perturbation (treatment with ~3000 small-molecule compounds), and therefore provides the cell-autonomous changes in the gene expression induced by the perturbagen. We analyzed the correlation between the transcriptional profile of the tissues in our study and the profile induced by SFN treatment or NRF2 overexpression in the CMap database. While liver and skeletal muscle had an RNAseq profile that highly correlated with the transcriptional signature of NRF2 overexpression, skeletal muscle was the only tissue that also correlated with the SFN-induced gene expression profile (*Figure 6C*). A similar analysis conducted on the RNAseq results obtained from the tissues of NRF2 KO mice following vehicle or SFN treatments yielded no significant correlation for any of the tissues analyzed with both SFN and NRF2 overexpression signatures in CMap (*Figure 6C*). Furthermore, in agreement with our qPCR analysis (*Figure 6A*), RNAseq results from wild-type mice indicated a significant upregulation of the NRF2 target gene *Hmox1* only in the skeletal muscle (*Supplementary file 1*). We also conducted an enrichment analysis to identify potential transcription factors for the significantly upregulated genes (*Kuleshov et al., 2016*). NRF2 was identified as the most significant transcription factor only for the skeletal muscle

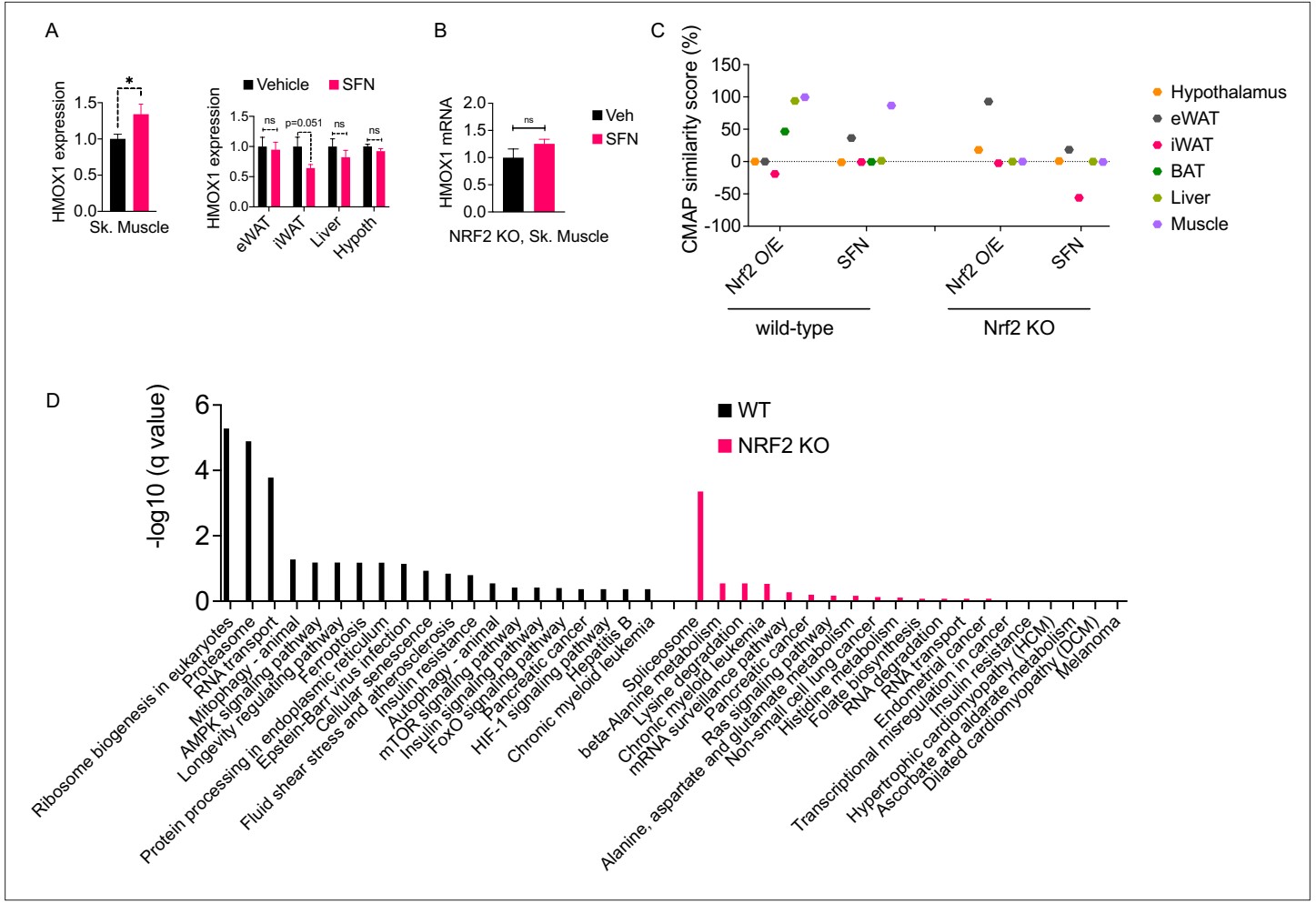

**Figure 6.** The molecular signature of sulforaphane (SFN) action in the obese mice. (**A**) The expression of NRF2-target gene *Hmox1* in the gastrocnemius skeletal muscle (left panel), and eWAT, inguinal white adipose tissue (iWAT), liver, the hypothalamus (right panel) measured by qPCR of diet-induced obese (DIO) wild-type mice (n = 7 per group) following 1 week of vehicle or 5 mg/kg SFN treatments. *p<0.05, calculated by Student's *t*-test. (**B**) *Hmox1* mRNA expression measured by qPCR in the skeletal muscle of DIO NRF2 KO mice (n = 8 per group) following i.p. vehicle or 5 mg/kg SFN treatments. (**C, D**) RNAseq was conducted on DIO wild-type NRF2 KO mouse tissues following vehicle or SFN treatment (n = 5 per tissue, per treatment). (**C**) Connectivity Map (CMap) correlation score of the SFN gene expression signature from each tissue. (**D**) KEGG enrichment pathway analysis of skeletal muscle based on the RNAseq results from WT DIO (left panel) and NRF2 KO DIO (right panel) mice.

The online version of this article includes the following figure supplement(s) for figure 6:

**Figure supplement 1.** RNAseq results of the vehicle vs. sulforaphane (SFN)-treated wild-type diet-induced obese (DIO) mice.

**Figure supplement 2.** Pathway enrichment analyses of bulk RNA sequencing results indicating significantly altered genes following SFN treatments in tissues of WT DIO and NRF2 KO DIO mice.

**Figure supplement 3.** Pathway enrichment analyses of bulk RNA sequencing results indicating significantly altered genes following SFN treatments in tissues of WT DIO and NRF2 KO DIO mice.

**Figure supplement 4.** Pathway enrichment analyses of bulk RNA sequencing results indicating significantly altered genes following SFN treatments in tissues of WT DIO and NRF2 KO DIO mice.

**Figure supplement 5.** Pathway enrichment analyses of bulk RNA sequencing results indicating significantly altered genes following SFN treatments in tissues of WT DIO and NRF2 KO DIO mice.

**Figure supplement 6.** Pathway enrichment analyses of bulk RNA sequencing results indicating significantly altered genes following SFN treatments in tissues of WT DIO and NRF2 KO DIO mice.

**Figure supplement 7.** Pathway enrichment analyses of bulk RNA sequencing results indicating significantly altered genes following SFN treatments in tissues of WT DIO and NRF2 KO DIO mice.

*Figure 6 continued on next page*

*Figure 6 continued*

**Figure supplement 8.** Pathway enrichment analyses of bulk RNA sequencing results indicating significantly altered genes following SFN treatments in tissues of WT DIO and NRF2 KO DIO mice.

**Figure supplement 9.** Pathway enrichment analyses of bulk RNA sequencing results indicating significantly altered genes following SFN treatments in tissues of WT DIO and NRF2 KO DIO mice.

**Figure supplement 10.** Pathway enrichment analyses of bulk RNA sequencing results indicating significantly altered genes following SFN treatments in tissues of WT DIO and NRF2 KO DIO mice.

**Figure supplement 11.** Pathway enrichment analyses of bulk RNA sequencing results indicating significantly altered genes following SFN treatments in tissues of WT DIO and NRF2 KO DIO mice.

**Figure supplement 12.** Pathway enrichment analyses of bulk RNA sequencing results indicating significantly altered genes following SFN treatments in tissues of WT DIO and NRF2 KO DIO mice.

**Figure supplement 13.** Pathway enrichment analyses of bulk RNA sequencing results indicating significantly altered genes following SFN treatments in tissues of WT DIO and NRF2 KO DIO mice.

**Figure supplement 14.** Pathway enrichment analyses of bulk RNA sequencing results indicating significantly altered genes following SFN treatments in tissues of WT DIO and NRF2 KO DIO mice.

**Figure supplement 15.** Pathway enrichment analyses of bulk RNA sequencing results indicating significantly altered genes following SFN treatments in tissues of WT DIO and NRF2 KO DIO mice.

**Figure supplement 16.** Pathway enrichment analyses of bulk RNA sequencing results indicating significantly altered genes following SFN treatments in tissues of WT DIO and NRF2 KO DIO mice.

**Figure supplement 17.** Pathway enrichment analyses of bulk RNA sequencing results indicating significantly altered genes following SFN treatments in tissues of WT DIO and NRF2 KO DIO mice.

**Figure supplement 18.** Pathway enrichment analyses of bulk RNA sequencing results indicating significantly altered genes following SFN treatments in tissues of WT DIO and NRF2 KO DIO mice.

**Figure supplement 19.** Pathway enrichment analyses of bulk RNA sequencing results indicating significantly altered genes following SFN treatments in tissues of WT DIO and NRF2 KO DIO mice.

**Figure supplement 20.** Pathway enrichment analyses of bulk RNA sequencing results indicating significantly altered genes following SFN treatments in tissues of WT DIO and NRF2 KO DIO mice.

**Figure supplement 21.** Pathway enrichment analyses of bulk RNA sequencing results indicating significantly altered genes following SFN treatments in tissues of WT DIO and NRF2 KO DIO mice.

**Figure supplement 22.** Pathway enrichment analyses of bulk RNA sequencing results indicating significantly altered genes following SFN treatments in tissues of WT DIO and NRF2 KO DIO mice.

**Figure supplement 23.** Pathway enrichment analyses of bulk RNA sequencing results indicating significantly altered genes following SFN treatments in tissues of WT DIO and NRF2 KO DIO mice.

**Figure supplement 24.** Pathway enrichment analyses of bulk RNA sequencing results indicating significantly altered genes following SFN treatments in tissues of WT DIO and NRF2 KO DIO mice.

**Figure supplement 25.** Pathway enrichment analyses of bulk RNA sequencing results indicating significantly altered genes following SFN treatments in tissues of WT DIO and NRF2 KO DIO mice.

**Figure supplement 26.** Pathway enrichment analyses of bulk RNA sequencing results indicating significantly altered genes following SFN treatments in tissues of WT DIO and NRF2 KO DIO mice.

**Figure supplement 27.** Pathway enrichment analyses of bulk RNA sequencing results indicating significantly altered genes following SFN treatments in tissues of WT DIO and NRF2 KO DIO mice.

**Figure supplement 28.** Pathway enrichment analyses of bulk RNA sequencing results indicating significantly altered genes following SFN treatments in tissues of WT DIO and NRF2 KO DIO mice.

**Figure supplement 29.** Pathway enrichment analyses of bulk RNA sequencing results indicating significantly altered genes following SFN treatments in tissues of WT DIO and NRF2 KO DIO mice.

**Figure supplement 30.** Pathway enrichment analyses of bulk RNA sequencing results indicating significantly altered genes following SFN treatments in tissues of WT DIO and NRF2 KO DIO mice.

**Figure supplement 31.** Pathway enrichment analyses of bulk RNA sequencing results indicating significantly altered genes following SFN treatments in tissues of WT DIO and NRF2 KO DIO mice.

**Figure supplement 32.** Pathway enrichment analyses of bulk RNA sequencing results indicating significantly altered genes following SFN treatments in tissues of WT DIO mice.

**Figure supplement 33.** Pathway enrichment analyses of bulk RNA sequencing results indicating significantly altered genes following SFN treatments in tissues of WT DIO mice.

*Figure 6 continued*

**Figure supplement 34.** Pathway enrichment analyses of bulk RNA sequencing results indicating significantly altered genes following SFN treatments in tissues of WT DIO mice.

**Figure supplement 35.** Pathway enrichment analyses of bulk RNA sequencing results indicating significantly altered genes following SFN treatments in tissues of WT DIO mice.

**Figure supplement 36.** Pathway enrichment analyses of bulk RNA sequencing results indicating significantly altered genes following SFN treatments in tissues of WT DIO mice.

**Figure supplement 37.** Pathway enrichment analyses of bulk RNA sequencing results indicating significantly altered genes following SFN treatments in tissues of WT DIO mice.

gene set in wild-type mice (*Supplementary file 3*), providing further support for the skeletal muscle being a primary target tissue of SFN action. Interestingly, RNAseq results from NRF2 KO mice failed to show the SFN-induced upregulation of *Hmox1* in the skeletal muscle (*Supplementary file 2*). We thus conducted a KEGG pathway analysis of skeletal muscle transcriptome and identified ribosome biogenesis, proteasome pathway, the RNA transport, and mitophagy as the significantly upregulated pathways in SFN-treated wild-type mice. The KEGG pathway analysis of samples from NRF2 KO animals identified only the splicesome pathway as significantly upregulated following SFN treatment (*Figure 6D*, *Supplementary file 4*).

In addition to HMOX1, the RNAseq results from wild-type mice indicate upregulation of other genes involved in OSR capacity in SFN-treated mice in the skeletal muscle. *Gclm* (glutamate-cysteine ligase), the first rate-limiting enzyme of glutathione synthesis, *Srxn1* (Sulfiredoxin), an oxidoreductase that reduces cysteine-sulfinate, and GSR (glutathione-disulfide reductase), a central antioxidant enzyme, which reduces oxidized glutathione disulfide (*Gssg*) to the sulfhydryl form glutathione (*Gsh*), and *Txnrd1* (thioredoxin reductase 1), which reduces thioredoxins, are all upregulated in the skeletal muscle of the SFN-treated DIO wild-type animals (*Supplementary file 1*).

## Discussion

In this study, we use various mouse models to present evidence that the organosulfur natural compound SFN reverses diet-induced obesity. The antiobesity potential of SFN requires both a functional leptin receptor signaling as well as hyperleptinemia. SFN does not reduce the body weight or food intake of standard chow-fed wild-type lean, *Lepr^db/db^*, or leptin-deficient *Lep^ob/ob^* mice. However, when coadministered with exogenous leptin, SFN induces an anorectic response in standard chow-fed lean and *Lep^ob/ob^*, but not *Lepr^db/db^* mice, suggesting that the weight-reducing effect of the compound requires circulating leptin and functional leptin receptor signaling. Thus, SFN-induced weight loss and anorexia was observed only in DIO wild-type mice. We did not observe a decrease in the food intake of the lean animals when SFN administration started upon switching to HFD, at which point the mice did not have hyperleptinemia. Under such conditions of relatively lower leptin levels compared to hyperleptinemia observed in DIO mice, higher SFN concentrations were required to potentiate the anorexic effects of leptin. These findings are consistent with the requirement of leptin receptor signaling and relatively high serum leptin levels for the antiobesity action of SFN.

### SFN-treated mice lose fat mass, but not lean mass

We observed that SFN suppressed food intake in wild-type DIO mice. We postulate that the decreased energy intake contributes to the observed weight loss. We also investigated the effects of SFN on EE in DIO mice feeding ad lib on HFD. We did not observe increased EE in these mice; however, SFN prevented the calorie restriction-induced suppression of EE, suggesting that SFN may in fact activate mechanisms underlying promotion of EE. Additionally, SFN treatment significantly lowered RER, indicating increased fat utilization/fatty acid oxidation as an energy source, in agreement with the decreased fat mass of the SFN-treated animals observed in body composition analysis. SFN suppressed the expression of the fatty acid synthesis genes in the liver and white adipose tissues, consistent with the observation that SFN-treated mice lose significant fat mass. Locomotor activity in SFN-treated animals was not significantly different compared to the vehicle-treated counterparts, ruling out that increased physical activity contributed to decreased adiposity. This finding also suggests that SFN does not induce any apparent toxicity or aversive effects. In support of this

notion, SFN treatment did not decrease the food intake of lean mice on either standard chow or HFD, further supporting a lack of SFN-induced taste aversion in mice, in agreement with reports from human subjects (*Bahadoran et al., 2011*).

## Mechanisms underlying the metabolic effects of SFN

NRF2 activation by small-molecule compounds has been suggested to protect from diet-induced obesity (*Shin et al., 2009*). However, NRF2 KO mice gain less weight on HFD compared to wild-type animals (*Chan et al., 1996*; *Zhang et al., 2016*). While such discrepancy between pharmacology versus genetics is not uncommon (*Minikel et al., 2020*), we hypothesized that SFN does not exert NRF2 activation across all tissues. While central NRF2 activation through genetic approaches confers partial protection against weight gain on HFD (*Yagishita et al., 2017*), central administration of SFN failed to reduce body weight or food intake in DIO mice in our study, nor did SFN directly activate leptin receptor signaling as measured by pSTAT3 in LepRb-expressing hypothalamic N1 cells, suggesting that the effect of SFN on leptin sensitivity may be mediated through two sites of action, where SFN primarily acting in peripheral tissue(s) communicates indirectly to the brain to activate leptin receptor signaling. In this regard, an important question remains as to how peripheral SFN action is coupled to central leptin sensitization that ultimately leads to weight loss.

## Investigating the SFN's mechanism of action by using RNAseq

The transcriptional profiling of several tissues following SFN treatment indicated a significant induction of NRF2 activity specifically in the skeletal muscle, supporting a role of the muscle tissue as a primary target site of SFN action. Most of the effects we detected in the skeletal muscle of wild-type mice were absent in NRF2 KO mice. In support of this model, NRF2 activation through skeletal muscle-specific ablation of KEAP1 protects from the development of diet-induced obesity and metabolic syndrome in wild-type mice with a significantly compromised effect in $Lepr^{db/db}$ mice (*Uruno et al., 2016*; *Matzinger et al., 2018*), emphasizing the cell-nonautonomous connection between muscle NRF2 action and central leptin signaling. While we cannot rule out the potential contribution of other tissues not included in our analysis to SFN-induced phenotypes, it is tempting to speculate that direct SFN action on myocytes activates a transcriptional program that leads to central leptin sensitization, potentially through a systemic secreted factor.

We detected increased expression of BAT-marker genes, *Ucp1*, *Elovl3*, and *Cidea,* in the iWAT of NRF2 knockout mice, but not wild-type mice, in our RNAseq results obtained following SFN treatment. SFN exerts NRF2-independent actions such as inhibiting mTOR pathway through effects on HDAC6/AKT (*Zhang et al., 2021*). Furthermore, adipose-specific ablation of mTOR activity has been shown to induce the browning of WAT (*Polak et al., 2008*; *Zhang et al., 2018*), and thus could potentially account for the NRF2-independent effect of SFN on this phenotype. However, when we directly treated the iWAT explants with SFN, we did not detect increased expression of either brown or beige adipose marker genes. A former study suggested that the SFN precursor glucoraphanin increased browning of the iWAT in an NRF2-dependent manner (*Nagata et al., 2017*). While the underlying causes of the discrepancy between our results and this former study remain unclear to us, differences in drug doses, treatment periods, or mouse strains could possibly be considered. While NRF2 activation is proposed to directly induce *Ucp1* expression (*Chang et al., 2021*), fat-specific NRF2 KO mice had increased *Ucp1* expression in the iWAT (*Chartoumpekis et al., 2018*), Accordingly, it is possible that SFN-induced suppression of food intake and subsequent weight loss is normally compensated by NRF2-dependent mechanisms, leading to the downregulation of BAT-related genes in iWAT. Removal of these compensatory mechanisms in the NRF2 KO mice would result in the upregulated expression of BAT-related thermogenic genes in the iWAT. Although the role of NRF2 in beiging requires further investigation, taken together, our results suggest that SFN has an NRF2-independent effect on iWAT browning. Furthermore, this effect plays only a marginal role in SFN-induced weight loss in light of the fact that the antiobesity effect of SFN is significantly compromised in NRF2 KO mice.

One of the notable findings in our RNAseq analysis was the relatively low number of changes detected in the hypothalamic transcriptome. The hypothalamus is composed of more than 40 cell types with distinct transcriptional signatures (*Chen et al., 2017*). For example, POMC and AgRP neurons, which form the melanocortin circuitry that mediates approximately half of leptin action, represent only a minor fraction of the diverse hypothalamic cell populations. This heterogeneity is

further enriched within defined cell populations such that the leptin receptor-expressing hypothalamic cells represent at least 25 different neuronal and non-neuronal cell clusters (*Kakava-Georgiadou and Severens, 2020*). It is likely that transcriptional changes in distinct, chemically defined neuronal populations cannot be detected by total hypothalamic transcriptional analysis. Future studies on the transcriptional profile of leptin receptor-positive cell populations in SFN-treated mice will help uncover both the nature of the peripheral signals potentiating leptin signaling and the molecular biology of leptin resistance.

The SFN-induced increase in antioxidant capacity has been reported as a result of NRF2 activation of its downstream genes. Increased OSR capacity improves glucose tolerance and insulin sensitivity (*Kopprasch et al., 2016*); however, the role of peripheral OSR on the central leptin receptor signaling and body weight regulation is unclear. Therefore, an important question remains as to how stimulation of NRF2 activity/OSR in skeletal muscle can improve leptin receptor signaling and trigger weight loss in SFN-treated mice. Future studies will assess the antiobesity and leptin-sensitizing effects of SFN on muscle-specific NRF2 KO mice.

## Reducing body weight in an anabolic state

KEAP1 is the main negative regulator of NRF2 stability and therefore activity. KEAP1 deficiency in mice was proposed to induce a metabolic state mimicking fasting (*Knatko et al., 2020*), in large part because the resultant NRF2 activation leads to fatty acid oxidation. Our data also supports this state of fasting in the liver and adipose tissue in regard to fat metabolism. RNAseq results indicated downregulation of genes involved in fatty acid synthesis, acetyl-CoA, and oxaloacetate formation including *Fasn*, *Scd1*, *Acly*, and *Me1* in liver, BAT, and eWAT, suggesting decreased lipogenesis and cholesterogenesis. These findings are supported with GSEA of RNAseq results confirming significant downregulation of cholesterol, fatty acyl-CoA, and triglyceride biosynthesis in eWAT and liver (*Figure 1—figure supplement 1*, *Figure 5—figure supplement 2*, *Figure 6—figure supplements 8–13; 20–25*). However, this catabolic phenotype in fat metabolism, which parallels the fasted state, is not accompanied by protein catabolism as we further discuss below.

SFN activates multiple cellular pathways, directly and indirectly, and generates a complex molecular profile. For example, SFN-induced reduction of food intake and fat mass could potentially result in an energy deficit and activation of metabolic pathways that conserve energy, leading to an overall catabolic output. On the other hand, SFN and its precursor glucoraphanin improve glucose homeostasis and increase insulin action (*Figure 1G and H*; *Axelsson et al., 2017*; *Xu et al., 2018c*), which activates the major anabolic pathways that consume energy. Furthermore, SFN was proposed to inhibit the action of mTOR, an insulin signaling downstream molecule and major anabolic regulator, which might augment the energy-preserving pathways in key metabolic tissues. We thus decided to investigate further in-depth mechanisms of action of SFN using our large volume of RNAseq data by conducting pathway analyses to dissect the network-level changes induced by SFN in each tissue (*Figure 6C and D*, *Figure 5—figure supplement 3B*, *Figure 6—figure supplements 2–37* ). Analysis of our RNAseq results indicates that the SFN treatment results in concurrent changes in multiple crucial molecular pathways. Surprisingly, despite weight loss and decreased food intake induced by the SFN treatment, the RNAseq results demonstrated that SFN caused a striking upregulation of the most anabolic pathways in the high-energy-demanding tissues including the BAT, skeletal muscle, and liver (*Figure 6D*, *Figure 6—figure supplement 2*, *Figure 6—figure supplements 2–13; 32–37*). We observed robust upregulation of protein synthesis and ribosome biogenesis pathways supported by upregulation of genes such as eIF2 downstream pathway, the housekeeping ATP-dependent peptidase CLPP, and numerous ribosomal proteins. Markers of cell proliferation such as myc downstream genes were upregulated, and tumor-suppressing genes such as *Trim24*, *Trp53*, and *Rb1* had decreased expression. Additionally, we observed upregulation in cytoskeletal assembly and cell-cell adhesion molecule genes supported by upregulation of alpha-catenin downstream genes, N−glycan biosynthesis genes, and *Psmb11* and *Psmb7*. Increased ribosome biogenesis and overall markers of proteostasis are in agreement with preservation of the lean mass and EE of the DIO mice despite decreased food intake during SFN treatment. In line with these findings, SFN-treated mice were reported to have enhanced running capacity with attenuated muscle fatigue (*Oh et al., 2017*). Thus, overall metabolic profile induced by SFN is not a simple mimetic of fasting or associated with the catabolic state anticipated uniformly across the tissues due to caloric restriction.

One of the hallmarks of obesity is the chronic low-grade inflammation in several central and peripheral tissues (**Weisberg et al., 2003**; **Wellen and Hotamisligil, 2003**). The anti-inflammatory effect of NRF2 activation has been studied in the context of metabolic syndrome as well as during classical inflammation induced by pathogens such as viruses including SARS-COVID-19 (**Olagnier et al., 2020**). Furthermore, macrophage infiltration into the adipose tissue and their polarization towards a pro-inflammatory profile is causally linked to the etiology of obesity-induced insulin resistance (**Lee and Lam, 2019**). In agreement with these, the expression of the anti-inflammatory M2 macrophage marker genes *Chil3* and *Arg1* was upregulated (6.2- and 5.9-fold over vehicle for *Agr1* and *Chil3*, respectively, p<0.001) in the white adipose tissue of SFN-treated mice. Furthermore, expression of the anti-inflammatory cytokine IL10 is also increased in the eWAT by SFN (**Supplementary file 1**). However, it is worth noting that the same dose of SFN that induces potent weight loss in DIO mice is not effective to reduce either the body weight or improve the glucose homeostasis in *Lep^{ob/ob}* or *Lepr^{db/db}* mice. Therefore, while these findings suggest that SFN could contribute to the improved metabolic profile of the animals by decreased inflammatory tone, we propose that the anti-inflammatory function of NRF2 activation is not the major driver of SFN-induced weight loss.

## Therapeutic potential of SFN

Since SFN is acid labile and sensitive to heat, glucoraphanin, a more stable and inert precursor of SFN, is taken orally by humans as a food supplement. Glucoraphanin exists in high concentrations in the aqueous broccoli sprouts extract (BSE). Myrosinase, a β-thioglucosidase, converts glucoraphanin to SFN. This enzyme exists in plants as well as the mammalian microbiome (**Shapiro et al., 1998**; **Fahey et al., 2015**). With plant tissue degradation during chewing, myrosinase comes in contact with glucoraphanin that converts it to SFN. The mammalian cells do not produce myrosinases and the conversion of glucoraphanin to SFN occurs by the bacterial microflora of the gastrointestinal tract (**Shapiro et al., 1998**; **Yagishita et al., 2019**; **Bouranis et al., 2021**). Meta-analysis of published studies using mice indicates a close proximity of dose ranges for the efficacy of SFN on various pharmacological targets by i.p. and oral administrations, suggesting an excellent oral bioavailability. These studies have utilized a wide range of effective doses of SFN with the medians of 175 and 113 µmol/kg of body weight by oral and i.p. administration, respectively (**Yagishita et al., 2019**). In this study, we used the dose 5–10 mg/kg (30–60 µmol/kg) SFN to treat DIO mice to induce weight loss, clearly in the lower range of previously published studies. Human studies, however, report the efficacy dose ranges of SFN to be significantly (around 10-fold) lower than in mice, even after correction for allometric scaling. The lower efficacy range was reported as <0.5 µmol/kg and the median human dose around 300 µmol (4 µmol/kg) in human studies (**Shapiro et al., 1998**; **Yagishita et al., 2019**). These discrepancies could be due to greater clearance in rodents compared to humans per body weight (**Yagishita et al., 2019**). Furthermore, few rodent studies have included a dose-response data, with the majority of the animal study doses exceeding the highest doses of SFN used in humans, even after accounting for allometric scaling between rodents and humans. Thus, the greater than 4-log spread of doses used in mice appears not to be based on a dose-response analysis and optimization of translational science (**Yagishita et al., 2019**). Following oral administration, SFN and its metabolites were detected in various tissues at 2 and 6 hr, and a dose-dependent increase in its concentration was also observed in most tissues (**Clarke et al., 2011**). In humans, safety studies have shown that BSE at amounts corresponding to 50–400 µmol SFN daily is well tolerated without clinically significant adverse effects (**Kensler et al., 2005**; **Shapiro et al., 2006**; **Singh et al., 2014**). The toxicity dose (LD50) of SFN is estimated to be about 10-fold higher than the median dose reported for efficacy outcomes in mice (213 mg/kg i.p.; 1203 µmol/kg) (**Yagishita et al., 2019**). In clinical trials to investigate the effects of SFN on glucose control in T2D patients, BSE containing 150 µmol SFN per dose has been administered to humans. This dose (~3.3 mg/kg), corresponding to one-third of the dose per body surface area compared with the animal experiments (at 10 mg/kg), was well tolerated in clinical safety studies (**Axelsson et al., 2017**) and significantly reduced the blood glucose and glycated hemoglobin levels in obese diabetic patients without affecting the body weight (**Bahadoran et al., 2011**; **Cramer and Jeffery, 2011**; **Kikuchi et al., 2015**; **Axelsson et al., 2017**). While obese humans develop hyperleptinemia in a similar manner to diet-induced rodents, it is possible that the dose and bioavailability of SFN to various tissues are also important parameters to be considered as an antiobesity agent. Numerous clinical studies investigating SFN's potential beneficial effects in a wide range of diseases

including cancers (prostate, bladder, colorectal, breast, lung), psychiatric diseases (schizophrenia, autism, depression), inflammation (COVID-19, osteoarthritis), metabolic syndrome (DMII, obesity), and aging (https://clinicaltrials.gov) will ultimately uncover whether the findings we report here are to be translated to human health.

## Materials and methods

### Animals

The Institutional Animal Care and Use Committees (IACUC) at the University of Michigan (protocol number: PRO00007712), Vanderbilt University (protocol numbers: M1700112 and M/10/358), and Qatar University (protocol number: QU-IACUC 1-62019-1) approved the experimental protocols and euthanasia procedures used in this study. Animals were housed at a 12 hr dark/light cycle, temperature- and humidity-controlled rooms. Mice were purchased from the Jackson Laboratory. HDAC6 KO mice were provided by Timothy McKinsey (University of Colorado). Generation of the HDAC6 KO mice was described in *Demos-Davies and Ferguson, 2014* and *Gao et al., 2007*. Mice were fed either standard chow or HFD (60 kcal% fat, Research Diets) and had free access to food and water unless specified. For the CD1 mice and NRF2 KO mice, the following diets were used: regular diet: U8409G10R SAFA, France, HFD (HF260; U8978P Version 0019, SAFE, France).

Body composition was analyzed with Bruker's minispec LF50 Body Composition Analyzer (Bruker) when indicated.

### Leptin food intake studies

Lean wild-type mice were fasted for 25 hr. SFN was injected at 0, 16, and 24 hr of fasting. Leptin was dissolved in sterile PBS and i.p. injected 30 min prior to start of refeeding. Food intake was measured at 3, 6, 16, and 24 hr, and body weight was measured 16 and 24 hr after refeeding. *Lep^{ob/ob}* mice were pretreated with SFN or vehicle for three consecutive days, and then continued to receive vehicle or SFN with either PBS or leptin. Injections were done within 1 hr prior to start of dark cycle.

### Cell culture

MEF cells were cultured in high-glucose DMEM (Gibco) supplemented with 10% FBS and penicillin/ streptomycin. Tubastatin A HCl and SFN were dissolved in DMSO and added to the culture medium for 24 hr.

### Drugs and reagents

Tubastatin A HCl and SFN were purchased from different vendors (APExBIO, Selleckchem, AdooQ Bioscience, and Biopurify Phytochemicals). SFN was diluted in a solution of 50% PEG, 30% PBS, 20% DMSO, or only DMSO for animal studies. Recombinant mouse leptin was from A. F. Parlow (National Hormone and Peptide Program, Torrance, CA).

### Drug treatments

SFN was administered within 1 hr before dark cycle by daily i.p. injections unless specified otherwise. For i.p. injections, drugs or vehicle was injected at 25 µl volumes per animal. For lateral ventricle infusions, SFN was dissolved in DMSO as vehicle and was infused in 500 nl at indicated doses once a day prior to dark cycle.

### Lateral ventricle cannulation

Under isoflurane anesthesia, mice were stereotaxically implanted with a stainless steel cannula (Plastics One, VA) into their right lateral ventricle at the following coordinates with respect to bregma: lateral: 1.00 mm; anteroposterior: –0.460 mm; ventral: –2.20 mm. Positive cannulation was verified by measurement of water intake in response to ICV injection of angiotensin II (Sigma, MO).

### Glucose tolerance test and leptin ELISA

For GTT, mice were fasted overnight. In the morning, mice were injected 1 g/kg dextrose intraperitoneally. Blood glucose was measured from tail vein at 0, 15, 30, 60, 90, and 120 min after glucose injections. Leptin concentrations were measured using mouse plasma according to the manufacturer's

instructions. Blood was collected in heparinized vials and centrifuged for 60 min at 3000 rpm at 4°C. Plasma was collected and stored in –80°C until further processing.

## Western blot

For Western blot analysis, cells or tissues were lysed in RIPA buffer (50 mM TRIS pH 7.50, 25 mM NaF, 100 mM NaCl, 5 mM EDTA, 0.1% SDS, 1% TritonX-100) supplemented with protease and phosphatase inhibitors, 20 mM nicotinamide, and 20 µM vorinostat. Equal amounts of total lysates were separated on 4–15% SDS-PAGE gels (Bio-Rad), transferred to PVDF membranes (Millipore), and probed with indicated antibodies. Blots were washed with PBS/T (0.1% Tween-20 in PBS) and either developed (for GAPDH-HRP) or probed with HRP-conjugated secondary antibodies and developed.

## Gene expression analysis (qRT-PCR) and RNAseq

Total RNA (1 µg) isolated (Trizol Reagent, Invitrogen) from frozen tissues was converted to cDNA (cDNA reverse transcription kit, Invitrogen) and used to screen expression levels of the listed genes. Reactions were amplified in an ABI Prism 7500 FAST sequence detector (Applied Biosystems), and acquired data were analyzed using the ΔΔCt method to determine the expression level of each transcript normalized to the expression level of the housekeeping genes (*Rplp0*, *Tbp*, and/or *Actb*). RNAseq was conducted by Novogene, Co. Ltd. (Beijing, China) using total RNA DNase treated on RNA purification columns (QIAGEN).

## Indirect calorimetry

A standard 12 hr light/dark cycle was maintained throughout the calorimetry studies. Mice, after acclimation to individual housing for at least 7 days, were placed in metabolic cages located in the Mouse Metabolic Phenotyping Center at Vanderbilt University in a temperature- and humidity-controlled housing room. EE measures were obtained using a computer-controlled indirect calorimetry system (Promethion, Sable Systems, Las Vegas, NV). The calorimetry system consists of 16 metabolic cages (identical to home cages with bedding) each equipped with water bottles and food hoppers connected to load cells for food and water intake monitoring, and all animals had ad libitum access to food and water throughout the study unless otherwise specified. Respiratory quotient (RQ) is calculated as the ratio of $CO_2$ production over $O_2$ consumption. EE is calculated using the Weir equation: kcal/hr = 60 * (0.003941 * $VO_2$ + 0.001106 * $VCO_2$) (**Weir, 1949**). Ambulatory activity is determined simultaneously every second with the collection of the calorimetry data. Ambulatory activity and position are detected with XYZ beam arrays (BXYZ-R, Sable Systems) with a beam spacing of 1.0 cm interpolated to a centroid resolution of 0.25 cm. Consecutive adjacent infrared beam breaks are counted and converted to distance, with a minimum movement threshold set at 1 cm. Data acquisition and instrument control were coordinated by MetaScreen v2.2.18, and the obtained raw data were processed using ExpeData v1.7.30 (Sable Systems) using an analysis script detailing all aspects of data transformation. The script is available on request from Sable Systems.

## Primer sequences

Ms-*Acaca*-Fwd (TAATGGGCTGCTTCTGTGACTC)
Ms-*Acaca*-Rev (CTCAATATCGCCATCAGTCTTG)
Ms-*Arg1*-Fwd (CTCCAAGCCAAAGTCCTTAGAG)
Ms-*Arg1*-Rev (AGGAGCTGTCATTAGGGACATC)
Ms-*Cd36*-Fwd (GTCTTCCCAATAAGCATGTCTCC)
Ms-*Cd36*-Rev (ACTTTGATGGCCTCAACCTG)
Ms-*Chil3*-Fwd (AATGATTCCTGCTCCTGTGG)
Ms-*Chil3*-Rev(GGGGCCAGGCTTCTATTCC)
Ms-*Fabp4*-Fwd (GGAGCTGGGTTAGGTATGGG)
Ms-*Fabp4*-Rev (GGAGCTGGGTTAGGTATGGG)
Ms-*Fasn*-Fwd (GGAGGTGGTGATAGCCGGTAT)
Ms-*Fasn*-Rev (TGGGTAATCCATAGAGCCCAG)
Ms-*G6pc*-Fwd (CGACTCGCTATCTCCAAGTGA)
Ms-*G6pc*-Rev (CGACTCGCTATCTCCAAGTGA)
Ms-*Gck*-Fwd (ATGGCTGTGGATACTACAAGGA)
Ms-*Gck*-Rev (TTCAGGCCACGGTCCATCT)

Ms-*Leptin*-Fwd (GAGACCCCTGTGCGGTTC)
Ms-*Leptin*-Rev (CTGCGTGTGTGAAATGTCATTG)
Ms-*Lpl*-Fwd (TGTGTCTTCAGGGGTCCTTAG)
Ms-*Lpl*-Rev (GGGAGTTTGGCTCCAGAGTTT)
Ms-*Me1*-Fwd (GTCGTGCATCTCTCACAGAAG)
Ms-*Me1*-Rev (TGAGGGCAGTTGGTTTTATCTTT)
Ms-*Pck1*-Fwd (CTGCATAACGGTCTGGACTTC)
Ms-*Pck1*-Rev (CTGCATAACGGTCTGGACTTC)
Ms *Ppargc1a*-Fwd (AGCCGTGACCACTGACAACGAG)
Ms *Ppargc1a*-Rev (AGCCGTGACCACTGACAACGAG)
Ms-*Pklr*-Fwd (TCAAGGCAGGGATGAACATTG)
Ms-*Pklr*-Rev (CACGGGTCTGTAGCTGAGTG)
Ms-*Ppara*-Fwd (AACATCGAGTGTCGAATATGTGG)
Ms-*Ppara*-Rev (AGCCGAATAGTTCGCCGAAAG)
Ms-*Pparg*-Fwd (TCGCTGATGCACTGCCTATG)
Ms-*Pparg*-Rev (GAGAGGTCCACAGAGCTGATT)
Ms-*Scd1*-Fwd (GCTGGAGTACGTCTGGAGGAA)
Ms-*Scd1*-Rev (TCCCGAAGAGGCAGGTGTAG)
Ms-*Ucp1*-Fwd (CCTGCCTCTCTCGGAAACAA)
Ms-*Ucp1*-Rev (CCTGCCTCTCTCGGAAACAA)
Ms-*Tbp*-Fwd (GAAGCTGCGGTACAATTCCAG)
Ms-*Tbp*-Rev (CCCCTTGTACCCTTCACCAAT)
Ms-*Actb*-Fwd(GGCTGTATTCCCCTCCATCG)
Ms-*Actb*-Rev (CCAGTTGGTAACAATGCCATGT)
Ms-*Rplp0*-Fwd (AGATTCGGGATATGCTGTTGGC)
Ms-*Rplp0*-Rev (TCGGGTCCTAGACCAGTGTTC)

## Acknowledgements

This study was supported by the Qatar National Research Foundation grant (NPRP9-351-3-075) to NMR, MG-L and IC, the NIDDK Pilot & Feasibility grant to MG-L, and MDRC Pilot and Feasibility Grant (NIH Grant P30-DK020572) to IC. The indirect calorimetry study was performed by the Vanderbilt Mouse Metabolic Phenotyping Center (DK059637 and 1S10RR028101-01).

## Additional information

### Funding

| Funder | Grant reference number | Author |
| --- | --- | --- |
| Qatar National Research Fund | NPRP9-351-3-075 | Masoud Ghamari-Langroudi Nasser M Rizk Işın Çakır |
| National Institute of Diabetes and Digestive and Kidney Diseases | Pilot & Feasibility grant | Masoud Ghamari-Langroudi |
| National Institutes of Health | Michigan Diabetes Research Center Pilot and Feasibility Grant P30-DK020572 | Işın Çakır |

The funders had no role in study design, data collection and interpretation, or the decision to submit the work for publication.

### Author contributions

Işın Çakır, Conceptualization, Data curation, Funding acquisition, Supervision, Writing – original draft, Writing – review and editing; Pauline Lining Pan, Data curation, Formal analysis, Writing – original draft, Writing – review and editing; Colleen K Hadley, Abdulrahman El-Gamal, Amina Fadel, Dina

Elsayegh, Omnia Mohamed, Data curation; Nasser M Rizk, Data curation, Funding acquisition, Supervision, Writing – original draft; Masoud Ghamari-Langroudi, Conceptualization, Data curation, Formal analysis, Funding acquisition, Project administration, Resources, Supervision, Writing – original draft

**Author ORCIDs**
Işın Çakır http://orcid.org/0000-0003-4293-7267
Pauline Lining Pan http://orcid.org/0000-0001-7961-6307
Nasser M Rizk http://orcid.org/0000-0002-6288-3609
Masoud Ghamari-Langroudi http://orcid.org/0000-0001-7222-8512

**Ethics**
The Institutional Animal Care and Use Committees (IACUC) at the University of Michigan (protocol number: PRO00007712), Vanderbilt University (protocol numbers: M1700112 and M/10/358), and Qatar University (protocol number: QU-IACUC 1-62019-1) approved the experimental protocols and euthanasia procedures used in this study.

**Decision letter and Author response**
Decision letter https://doi.org/10.7554/eLife.67368.sa1
Author response https://doi.org/10.7554/eLife.67368.sa2

---

## Additional files

**Supplementary files**
• Supplementary file 1. Profiles of global transcriptional changes obtained from bulk RNAseq indicate significantly altered genes following 1-week sulforaphane (SFN) treatment in skeletal muscle, liver, inguinal white adipose tissue (iWAT), epididymal white adipose tissue (eWAT), hypothalamus, and brown adipose tissue (BAT) of WT diet-induced obese (DIO) mice. The highest transcriptional changes were detected in the liver, and the least number of transcriptional changes were observed in the hypothalamus.

• Supplementary file 2. Profiles of global transcriptional changes obtained from bulk RNAseq indicate significantly altered genes following a 4-week sulforaphane (SFN) treatment in skeletal muscle, liver, inguinal white adipose tissue (iWAT), epididymal white adipose tissue (eWAT), hypothalamus, and brown adipose tissue (BAT) of NRF2 KO diet-induced obese (DIO) mice. Results failed to show the SFN-induced upregulation of *Hmox1* in the skeletal muscle of NRF2 KO mice.

• Supplementary file 3. Profiles of potential transcription factors for the significantly upregulated genes following 1-week sulforaphane (SFN) treatment in skeletal muscle, liver, inguinal white adipose tissue (iWAT), epididymal white adipose tissue (eWAT), hypothalamus, and brown adipose tissue (BAT) of WT diet-induced obese (DIO) mice. NRF2 is identified as the most significant transcription factor only for the skeletal muscle gene set in WT mice.

• Supplementary file 4. Profiles of KEGG, Reactome, and GO enrichment of pathway analyses indicating significantly altered genes following sulforaphane (SFN) treatments in skeletal muscle, liver, inguinal white adipose tissue (iWAT), epididymal white adipose tissue (eWAT) and the hypothalamus of WT diet-induced obese (DIO) and NRF2 KO DIO, mice, and the brown adipose tissue (BAT) of WT DIO mice. KEGG pathway analysis of skeletal muscle transcriptome identifies ribosome biogenesis, proteasome pathway, the RNA transport, and mitophagy as the significantly upregulated pathways in SFN-treated wild-type mice. The KEGG pathway analysis of muscle samples from NRF2 KO animals identified only the splicesome pathway as significantly upregulated following SFN treatments.

• Transparent reporting form

**Data availability**
Sequencing data have been deposited in GEO under accession codes GSE181477.

The following datasets were generated:

| Author(s) | Year | Dataset title | Dataset URL | Database and Identifier |
|---|---|---|---|---|
| Pauline Lining P, Colleen HK, Abdulrahman EG, Amina F, Dina E, Omnia M, Nasser R, Masoud GL | 2022 | Sulforaphane Reduces Obesity by Reversing Leptin Resistance | https://doi.org/10.5061/dryad.z34tmpgd0 | Dryad Digital Repository, 10.5061/dryad.z34tmpgd0 |
| Çakır I, Lining Pan P, Hadley CK, El-Gamal A, Fadel A, Elsayegh D, Fadel A, Rizk NM, Ghamari-Langroudi M | 2022 | Sulforaphane Reduces Obesity by Reversing Leptin Resistance | https://www.ncbi.nlm.nih.gov/geo/query/acc.cgi?acc=GSE181477 | NCBI Gene Expression Omnibus, GSE181477 |

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
