## [Decision Letter]

**Decision letter after peer review:**

Thank you for submitting your article "Sulforaphane Reduces Obesity by Reversing Leptin Resistance" for consideration by *eLife*. Your article has been reviewed by 2 peer reviewers, and the evaluation has been overseen by Ana Domingos as a Reviewing Editor and Carlos Isales as the Senior Editor. The following individual involved in review of your submission has agreed to reveal their identity: Qingchun Tong (Reviewer #2).

Essential revisions:

Before publication is granted, authors should edit the manuscript to address the major concerns, which are central for supporting the conclusions:

1. Uniformize across figures 1, 3, 4, 5 the units by which BW is expressed, and clarify whether the BW of Nrf2 KO is wild type under chow or HDF for the same period of treatment.

2. Clarify if total ambulatory activity unclear vehicle in Figure 2F, 2G, 2H is affected

3. Clarify Asterisks in Figure 2C and run 2-way ANOVA in Figure 4f.

4. Discuss what accounts for the protection of the obese phenotype in the HFD SFN group Figure 3D

5. Provide more details on when the mice were used for GTTs and CLAMS

6. Include a chow-VEH group in Figure s3 D, E. and F.

7. Briefly discuss SFN pharmacology relative to a nutraceutical dose.

*Reviewer #1:*

The current study examines pharmacological activation of NRF2 in reversing and preventing diet induced obesity in mice. The authors demonstrate that NRF2-dependent lowering of body weight does not occur in lean animals and requires leptin receptor/hyperleptinemia. They also suggest a primary action in skeletal muscle and upregulation of a transcriptional program promoting a protection against obesity. There is some confusion/concerns about inclusion/exclusion of data within graphs, assumptions generated from the data, and interpretation of physiological drivers of the phenotype observed.

Cakir et al., examines whether pharmacological activation of NRF2 reverses DIO in mice. They showed that NRF2-dependent lowering of body weight does not occur in lean animals and requires leptin receptor/hyperleptinemia. They also suggest a primary action in skeletal muscle and upregulation of a transcriptional program promoting a protection against obesity.

Figures 1, 3, 4, 5. The body weight data in response to SFN and leptin is presented at times as total body weight and at other times percent change or change in grams. This makes it extremely difficult to compare across the different manipulations. Either this needs to be consistently presented throughout the manuscript or all 3 analyses should be presented for each manipulation. For instance, Figure 1A shows a 15% decrease in body weight of DIO mice however the total body weight as presented in Figure 1C is not shown for Figure 1A. However, in Figure 4A the total body weight is shown without the percent change. It appears from the graph in Figure 4A that there is 0% body weight change in Nrf2KO, vehicle; while the Nrf2KO, SFN may have reduced 5% maybe even up to 10%. If there was a change, was this significant? Similar comparisons should be shown in Figures 1C, 4D, 5A, 5C, 5F, Supp Figure 1, Supp Figure 5.

The authors make an assumption that the glucose effects are dependent upon body weight (page 10). Were the GTT measurements obtained from mice of a similar body weight? Could the authors plot the blood glucose in the GTT experiments against weight to determine if theres a connection?

Also, an overnight fast prior to GTT is not physiologically relevant since this completely depletes glycogen stores and increased gluconeogenesis, adipose tissue lipolysis – its not a physiological relevant way to assess glucose metabolism.

Also, if the authors were to normalize to percent change for the GTT, would they have the same conclusion?

Figure 2F, 2G, 2H unclear if vehicle does or does not affect activity. Is total ambulatory activity affected?

Figure 2C. Asterisks are confusing. What are the comparisons? In some, the asterisk is over the black bar, while in others they are over the red bars. In one, there is a double asterisk which is stacked vertically, while in others double and triple asterisks are horizontal.

Figure 3D. This presentation leads to the conclusion that there is no change in HFD and HFD SFN food intake. If so, then what accounts for the protection of the obese phenotype in the HFD SFN group? This would appear at odds with DIO mice (Figure 1 and 2). This should be discussed.

Figure 4. There appear to be issues with the controls in this figure. In figures 4A-4B, Nrf2KO mice receive vehicle and SFN. It seems the authors are missing vehicle and SFN groups for WT mice over this same period. In Figures 4D-4E, the authors present SFN treatment to WT and HDAC6KO mice. However, it appears they lack the vehicle control groups for these figures.

Figure 4F, presents WT vehicle/SFN; Nrf2KO Veh/SFN; and WT/HDAC6KO SFN – it appears they are missing the vehicle for HDAC6KO. Also, if these summary data are derived from Figure 4A-4E, are they comparable? In particular, Figure 4A-4C represents 27 days of treatment, while Figures 4D-4E represent 11 days of treatment. Overall it is very difficult to assess these data.

Figure 4f should be 2way anova

Figure 5E. it's surprising that leptin alone appears not to reduce cumulative food intake. However, this is not easily determined as the authors do not appear to have controlled for the effect of leptin. Previous studies have shown very rapid reductions (within 1d) in food intake in obob mice receiving as little as 1ug per day (PMID: 9421392). It would be useful to determine the effectiveness of leptin alone in these studies.

Figure 5g and 5h. Was the leptin level physiological in serum? Were these mice hyperleptinemic?

If SFN requires high leptin levels how does it work on lean mice which are fed HFD and administered SFN vs vehicle if they are starting with similar adiposity and presumably similar leptin levels?

*Reviewer #2:*

Cakir et al., examined the underlying mechanisms for the herb extract Sulforaphane (SFN) in reducing obesity. This study appears to be based on a large body of existing literature demonstrating a beneficial effect of SFN in body weight and metabolism but with an unclear mechanism. They first showed that SFN is highly selective on reducing HFD-induced obesity but has no obvious impact on chow-fed weight. Using a variety of relevant animal models, this study elegantly showed that the SFN effect is dependent on leptin signaling as it has no effects in both ob/ob an db/db animal models with deficient leptin function. Given the potential involvements of multiple downstream signaling pathways, the authors provided data that SFN is not effective in reducing HFD-induced obesity in NRF2 KO mice but is effective in HDAC6 deficient mice, arguing NRF is the main downstream mediator in reducing obesity. Intriguingly, additional data suggested that the SNF action is not mediated by hypothalamic neurons, where leptin resistance is thought to reside. The authors then performed RNAseq studies in various relevant peripheral tissues , trying to identify key underlying signaling pathways, with some data pointing to a potential involvement of muscle tissues in mediating the SFN effect. Overall, the study is very well designed and the results are clearly presented, and the manuscript is also well-written.

Strength:

1) As SFN has been extensively studied with demonstrated effects on reducing obesity but no unified underlying mechanisms, this study points to a clear mechanism with leptin action and NRF2, significantly extending the biology of SFN, and will have a significant impact on the field of obesity as well as its prevention and treatment.

2) Beautiful use of animal models with clean effects of SFN in the impact of obesity, strongly supporting the concept that the SFN effect is mediated by the leptin pathway.

3) The synergistic effect of leptin+SFN on reducing lean mouse body weight provides additional support for leptin action in mediating the SFN function.

4) The data on Nrf2 KO mice with no obvious effects on body weight strongly support that Nrf2 is the major downstream mediator of SFN.

Weakness:

1) One key concept is that the SFN effect on reducing obesity is mediated by its effect on activating the NRF2 signaling; however, NRF2 KO mice exhibit resistance to diet-induced obesity. As the authors have data from NRF2 KO mice, they didn't provide any data on these contradicting results.

2) This study will be benefited from providing more details on procedures including when or what ages the mice were subjected to GTTs and CLAMS measurements.

3) The RNAseq data in muscle/liver etc would be more informative in pointing to the underlying signaling pathways if the authors include the chow diet fed groups and/or Nrf2 KO groups.

This study is well executed and significant in the field of obesity research. The authors are encouraged to address the following specific concerns to clarify and strengthen the conclusion.

1) Please provide more details on when the mice were used for GTTs and CLAMS. This information is important for readers to assess whether this is an existing difference in body weight, which might cause secondary effects on GTTs and EE comparisons between the groups.

2) The issue with Nrf2 KO being resistant to DIO is concerning, which invites a slight possibility that SFN having no impact of Srf2KO may be due to the relatively lower obesity in these mice as compared to controls as SFN only produces less than 50% reduction on DIO. Can the authors plot the data on Nrf2KO and controls together in DIO with saline and SFN treatments to assess this possibility?

3) In Figure 3 D, E. and F, the chow-VEH group should also be included to allow assessment on the degree of SFN effects on these parameters.

4) In the RNAseq data, it is possible to include the groups of chow-fed and Nrf2 KO mice, which will provide much more information on specific changes that are relevant to obesity-reducing effects.

5) As SFN is a herb extract, the authors may need to discuss the doses used in this study and its relevance to the herb consumption.

---

## [Author Response]

Essential revisions:Before publication is granted, authors should edit the manuscript to address the major concerns, which are central for supporting the conclusions:1. Uniformize across figures 1, 3, 4, 5 the units by which BW is expressed, and clarify whether the BW of Nrf2 KO is wild type under chow or HDF for the same period of treatment.

The scales, ranges, and variables expressing body weight from various obese mouse models are now consistent across all figures for easier comparison.

2. Clarify if total ambulatory activity unclear vehicle in Figure 2F, 2G, 2H is affected

The summary results of statistical tests are now added to Figures 2F, 2G, and 2H to indicate that the ambulatory activity is not different between the vehicle and SFN groups.

3. Clarify Asterisks in Figure 2C and run 2-way ANOVA in Figure 4f.

The asterisk in Figure 2C is now fixed. Previous Figure 4F is revised to form the current Figure 4D. 2-way ANOVA has been conducted for proper statistics.

4. Discuss what accounts for the protection of the obese phenotype in the HFD SFN group Figure 3D

Results from metabolic chambers (Figure 2) show that SFN prevents the hypophagia-induced decrease in energy expenditure, suggesting that SFN induces energy expenditure. Therefore increased energy expenditure likely results in the attenuation of weight gain by SFN during the initial weeks of exposure to HFD of otherwise lean animals. The difference in the body weights of the HFD-Veh vs. HFD-SFN groups becomes significant (by 2-way ANOVA) after 2 weeks of HFD exposure.

SFN suppresses food intake in mice that are already obese and hyperleptinemic. Therefore in young, lean mice that have relatively low leptin levels, SFN did not suppress food intake during initial exposure of animals to HFD, suggesting that SFN does not induce a specific aversion against HFD. These results are consistent with the mechanism of action of SFN, which requires high leptin levels for SFN to suppress food intake.

Following 8 weeks of HFD exposure, the leptin levels in SFN-treated mice were significantly lower than vehicle-treated mice. We anticipate that as the leptin levels rise, SFN’s effect on food intake becomes more prominent. Accordingly, in the last week of the experiment (Figure 3D) the food intake of the SFN group started to show a tendency for a decrease, although the two-way ANOVA did not indicate statistical significance. Hence, we repeated this experiment with a slightly increased dose of SFN (15 mg/kg) where SFN significantly suppressed the cumulative food intake of the mice after two weeks of HFD exposure (Supplementary Figure 1L) suggesting that SFN action is dose dependent and slightly higher doses of SFN is required to suppress food intake when leptin levels are relatively low compared to DIO mice.

Our findings collectively suggest that SFN-induced leptin sensitization requires a potentially muscle-derived factor, which alone might be promoting energy expenditure, but in combination with hyperleptinemia elicits a potent anorectic weight-reducing effect. It is also possible that the mechanism of SFN-induced sensitization to leptin’s effect on food intake vs. energy expenditure might not be identical, however, this question requires identification of the precise mechanism of SFN-induced leptin sensitization. Neverthless, our results suggest that SFN attenuates weight gain on HFD in a dose-dependent manner through a mechanism that requires leptin signaling.

5. Provide more details on when the mice were used for GTTs and CLAMS

We conducted GTT on DIO mice following two weeks of SFN or vehicle treatments, at which point the SFNtreated mice had significant weight loss (Figures 1I, J). Because SFN induces weight loss in DIO mice, it is not trivial to attribute the improved glucose tolerance to the drug’s direct effect on glucose homesotasis or reduced adiposity, or both. To address this concern, we conducted GTT in *db/db* and *ob/ob* mice, which are resistant to the weight-reducing effects of SFN (Supplementary figure 4), where SFN did not improve the glucose tolerance. We also conducted GTT following 6-hr day-time fasting in DIO wild-type mice (Supplementary Figure 1 D, E), where SFN again improved glucose tolerance.

In the revised manuscript, we provide new data where SFN does not improve glucose tolerance in lean wildtype mice (Supplementary Figure 1I, J ). Furthermore, our new data show that SFN does not improve glucose tolerance in nrf2 KO mice (Figure 4G, H), either. Collectively these results suggest that, under the conditions we tested, the main effect of SFN on glucose metabolism is secondary to weight loss.

To conduct indirect calorimetry, mice were acclimated to single housing and daily handling for at least one week, and saline ip injections for three days prior to placing them into metabolic cages for the actual experiment. Mice started receiving a daily injection of SFN once they were in the metabolic cages. We routinely conduct indirect calorimetry in our facilities at Vanderbilt Mouse Metabolic Phenotype Core. The cage size and environment in the Promethion SABLE systems are similar to the animals’ home cage. Furthermore, we discard results from the first 48 hours of each run. We thus expect minimal effects of novelty-induced stress on our results.

6. Include a chow-VEH group in Figure s3 D, E. and F.

The figures have been revised to include the Chow-Veh group.

7. Briefly discuss SFN pharmacology relative to a nutraceutical dose.

The last paragraph in the discussion has been revised to include the relevant information.

Reviewer #1 (Recommendations for the authors):The current study examines pharmacological activation of NRF2 in reversing and preventing diet induced obesity in mice. The authors demonstrate that NRF2-dependent lowering of body weight does not occur in lean animals and requires leptin receptor/hyperleptinemia. They also suggest a primary action in skeletal muscle and upregulation of a transcriptional program promoting a protection against obesity. There is some confusion/concerns about inclusion/exclusion of data within graphs, assumptions generated from the data, and interpretation of physiological drivers of the phenotype observed.Cakir et al., examines whether pharmacological activation of NRF2 reverses DIO in mice. They showed that NRF2-dependent lowering of body weight does not occur in lean animals and requires leptin receptor/hyperleptinemia. They also suggest a primary action in skeletal muscle and upregulation of a transcriptional program promoting a protection against obesity.Figures 1, 3, 4, 5. The body weight data in response to SFN and leptin is presented at times as total body weight and at other times percent change or change in grams. This makes it extremely difficult to compare across the different manipulations. Either this needs to be consistently presented throughout the manuscript or all 3 analyses should be presented for each manipulation. For instance, Figure 1A shows a 15% decrease in body weight of DIO mice however the total body weight as presented in Figure 1C is not shown for Figure 1A. However, in Figure 4A the total body weight is shown without the percent change. It appears from the graph in Figure 4A that there is 0% body weight change in Nrf2KO, vehicle; while the Nrf2KO, SFN may have reduced 5% maybe even up to 10%. If there was a change, was this significant? Similar comparisons should be shown in Figures 1C, 4D, 5A, 5C, 5F, Supp Figure 1, Supp Figure 5.

The revised figures now show absolute body weights and percent changes in body weight for comparison for all cohorts.

– SFN-treatment does result in a small but significant change in the body weights of DIO nrf2KO mice. We observed that SFN leads to increased expression of brown adipose-related genes in inguinal WAT of nrf2KO mice. Through this nrf2-independent pathway of thermoregulation, SFN leads to increased expression of brown adipose related (“beiging”) genes that could potentially explain the observed SFN induced weight loss in nrf2KO mice (Supplementary Figure 3C-F).

It is important to note that a former study (Nagata et al., Diabetes. 2017 May;66(5):1222-1236.) had suggested that SFN protected against obesity by promoting “beiging” in an NRF2-dependent manner. Our results clearly indicate that SFN-induced beiging is NRF2-independent and contributes only marginally to the anti-obesity effect of SFN.

The authors make an assumption that the glucose effects are dependent upon body weight (page 10). Were the GTT measurements obtained from mice of a similar body weight? Could the authors plot the blood glucose in the GTT experiments against weight to determine if theres a connection?

– We conducted GTT on DIO mice following two weeks of SFN or vehicle treatments at which point the SFNtreated mice had significant weight loss. Because SFN induces weight loss in DIO mice, it is not trivial to attribute the improved glucose tolerance to the drug or reduced adiposity or both. To address these concerns, we conducted GTT in *db/db* and *ob/ob* mice, which are resistant to the weight reducing effects of SFN (Supplementary figure 4) where SFN did not improve the glucose tolerance.

Additionally, we now show in the revised manuscript that SFN does not improve glucose tolerance in lean wildtype mice (Supplementary Figure 1 I, J). Furthermore, our new data from nrf2KO mice shows that SFN does not improve glucose tolerance in nrf2KO mice (Figure 4 G, H), either. Collectively these results suggest that, under the conditions we tested, the main effect of SFN on glucose metabolism is secondary to weight loss.

Also, an overnight fast prior to GTT is not physiologically relevant since this completely depletes glycogen stores and increased gluconeogenesis, adipose tissue lipolysis – its not a physiological relevant way to assess glucose metabolism.

We now provide GTT results from DIO mice conducted after 6hr day-time fasting showing that SFN improves glucose tolerance (Supplementary Figure 1 D-G). These results support the results in Figure 1 where GTT was conducted after overnight fasting.

Also, if the authors were to normalize to percent change for the GTT, would they have the same conclusion?

We present percent change for the GTT (Supplementary Figure 1 F, G), where the difference between vehicle and SFN-treated mice is not significant.

Figure 2F, 2G, 2H unclear if vehicle does or does not affect activity. Is total ambulatory activity affected?

The statistics is now added to Figures 2F, 2G, and 2H to indicate that the ambulatory activity is not different between the vehicle and SFN groups.

Figure 2C. Asterisks are confusing. What are the comparisons? In some, the asterisk is over the black bar, while in others they are over the red bars. In one, there is a double asterisk which is stacked vertically, while in others double and triple asterisks are horizontal.

The figure has been revised, and the asterisks have been corrected.

Figure 3D. This presentation leads to the conclusion that there is no change in HFD and HFD SFN food intake. If so, then what accounts for the protection of the obese phenotype in the HFD SFN group? This would appear at odds with DIO mice (Figure 1 and 2). This should be discussed.

We did not observe a decrease in the food intake of the lean animals when SFN administration at 10mg/kg started upon switching to HFD. Increasing the SFN dose to 15 mg/kg led to a significant food intake suppression in lean mice placed on HFD after two weeks of treatment, pointing to a dose-response effect of the drug. However, the extent of inhibition of food intake by SFN in this experimental setting was still lower compared to its effect in the food intake of DIO mice. This is consistent with the mechanism of action of SFN, which requires high leptin levels to suppress food intake. As SFN potently suppressed food intake in mice that are obese, to begin with, i.e., DIO mice.

Please also see our response to editorial comment #5 above.

Figure 4. There appear to be issues with the controls in this figure. In figures 4A-4B, Nrf2KO mice receive vehicle and SFN. It seems the authors are missing vehicle and SFN groups for WT mice over this same period.

We now present in Figure 4C the body weight graphs for wild-type and nrf2 KO mice treated with vehicle or SFN for direct comparison.

In Figures 4D-4E, the authors present SFN treatment to WT and HDAC6KO mice. However, it appears they lack the vehicle control groups for these figures.

The revised graphs now include the acclimation period for both WT and hdac6 KO mice. SFN treatment leads to a significant decrease in the body weight of either genotype when compared to their body weights during the acclimation periods.

Figure 4F, presents WT vehicle/SFN; Nrf2KO Veh/SFN; and WT/HDAC6KO SFN – it appears they are missing the vehicle for HDAC6KO. Also, if these summary data are derived from Figure 4A-4E, are they comparable? In particular, Figure 4A-4C represents 27 days of treatment, while Figures 4D-4E represent 11 days of treatment. Overall it is very difficult to assess these data.

We have now revised this graph and removed the results from the hdac6 KO cohort; as the reviewer pointed out, the treatment time frames for the hdac6 KO cohorts were different than the other groups. In the revised graph, we present the summary results of WT and nrf2 KO mice treated with either vehicle or SFN in Figure 4D, and present hdac6 KO mice and WT littermates treated with SFN separately in Figure 4 I-K.

Figure 4f should be 2way anova

We now present two way ANOVA statistics on the revised Figure 4D

Figure 5E. it's surprising that leptin alone appears not to reduce cumulative food intake. However, this is not easily determined as the authors do not appear to have controlled for the effect of leptin. Previous studies have shown very rapid reductions (within 1d) in food intake in obob mice receiving as little as 1ug per day (PMID: 9421392). It would be useful to determine the effectiveness of leptin alone in these studies.

We now show in the revised Figure 5I that leptin decreases the food intake compared to the vehicle acclimation phase in both groups of mice. The reduction is more robust in the SFN group, suggesting the potentiation of leptin action. A similar trend is observed in the body weight of the mice where leptin induces a significant weight loss in both groups compared to their body weights during the acclimation phase. Leptin induced weight loss is more prominent when leptin is co-administered with SFN.

Figure 5g and 5h. Was the leptin level physiological in serum? Were these mice hyperleptinemic?

We have not measured the plasma leptin levels during these experiments. Based on the published half-life and dose response studies (Harris *et al.*, Endocrinology. 1998 Jan;139(1):8-19, and Burnett *et al.*, Int J Obes (Lond). 2017 Mar;41(3):355-359.), following the leptin injections we administered (200µg/kg, i.p.), we anticipate that there would be a transient hyperleptinemia in *ob/ob* mice.

If SFN requires high leptin levels how does it work on lean mice which are fed HFD and administered SFN vs vehicle if they are starting with similar adiposity and presumably similar leptin levels?

SFN suppresses weight gain in mice exposed to high fat diet. The difference in the body weight of the HFDVeh vs. HFD-SFN groups becomes significant after 2 weeks of HFD exposure at which point the animals start gaining adiposity and thus leptin levels start to increase.

We propose that SFN-induced leptin sensitization requires a potentially muscle-derived factor, which alone might be promoting energy expenditure, but in combination with hyperleptinemia elicits a potent weight reducing and anorectic response. This response is reminiscent of the catabolic signal circulating signal proposed for forced-feeding studies (Harris and Martin, 1990; White et al., 2010; Ravussin et al., 2014). It is also possible that the mechanism of SFN-induced sensitization to leptin’s effect on food intake vs. energy expenditure might may not be identical, however this question requires the identification of precise mechanism of leptin sensitization.

Please also see our response to question on Figure 3D above.

Reviewer #2 (Recommendations for the authors):[…]This study is well executed and significant in the field of obesity research. The authors are encouraged to address the following specific concerns to clarify and strengthen the conclusion.1) Please provide more details on when the mice were used for GTTs and CLAMS. This information is important for readers to assess whether this is an existing difference in body weight, which might cause secondary effects on GTTs and EE comparisons between the groups.

The revised manuscript now lists the details on GTTs and metabolic cage studies. Please also see our response to the editorial comment #6 above.

2) The issue with Nrf2 KO being resistant to DIO is concerning, which invites a slight possibility that SFN having no impact of Srf2KO may be due to the relatively lower obesity in these mice as compared to controls as SFN only produces less than 50% reduction on DIO. Can the authors plot the data on Nrf2KO and controls together in DIO with saline and SFN treatments to assess this possibility?

The phenotype of nrf2 KO mice on HFD has been controversial with some groups suggesting that the KOs are protected from diet-induced obesity (PMIDs: 21852674, 19698707, 20089859), while others showing the opposite (21262351, 23017736). Furthermore, keap1 hypomorphs, a genetic model of NRF2 activation, were shown to be protected from obesity in wild-type but not in *ob/ob* mice (PMIDs: 26701603, 22936178). This topic has been discussed extensively in a recent review (PMID: 32971975). It is now also discussed in the discussion page 17.

While we did not compare the growth curves of wild-type and nrf2 KO littermates, we used nrf2 KO mice that have been on HFD for 16 weeks, at which point the mice were obese and weighed over 45gm. This allowed us to avoid the potential problems the reviewer suggested; i.e, potentially low body weight of the nrf2 KOs confounding the interpretation of the findings.

As discussed elsewhere (e.g. below item#4), we observed that SFN leads to increased expression of brown adipose related genes in inguinal WAT of nrf2 KO mice. Through this NRF2-independent pathway of thermoregulation, SFN leads to increased expression of brown adipose related (“beiging”) genes that could potentially explain the observed SFN induced weight loss in nrf2 KO mice (Supplementary Figure 3 C-F). Consistent with these findings, the effect of SFN on body weight in obese nrf2 KO mice was only marginal (Figure 4 A-C).

– Per reviewer’s recommendation, we plotted the data from Nrf2 KO and wild-type mice in Figure 4C.

3) In Figure 3 D, E. and F, the chow-VEH group should also be included to allow assessment on the degree of SFN effects on these parameters.

The figures are now revised to include the chow-VEH groups (Figure 3E, F).

4) In the RNAseq data, it is possible to include the groups of chow-fed and Nrf2 KO mice, which will provide much more information on specific changes that are relevant to obesity-reducing effects.

We now provide bulk RNAseq results from DIO nrf2 KO mice treated with either vehicle or SFN (Supplementary Figure 3 C-F and Supplementary data 2). These results collectively suggest that most of the SFN-induced weight loss we observed in wild-type mice is NRF2-dependent.

Additionally, in contrast to a previous study (PMID: 28209760), these results also support existence of a SFNinduced NRF2-independent pathway, since SFN induces the expression of brown-adipose specific genes in inguinal white adipose tissue in nrf2 KO mice further showing that the beiging response to SFN is NRF2independent.

5) As SFN is a herb extract, the authors may need to discuss the doses used in this study and its relevance to the herb consumption.

The last paragraph in the discussion has been revised to include a discussion of the mouse and human doses used by previous publications.